# Deciphering the mechanism of processive ssDNA digestion by the Dna2-RPA ensemble

Jiangchuan Shen [1,5], Yiling Zhao[2,5], Nhung Tuyet Pham [3,5], Yuxi Li[1], Yixiang Zhang [4], Jonathan Trinidad [4], Grzegorz Ira[3], Zhi Qi [2✉] & Hengyao Niu [1✉]

Single-stranded DNA (ssDNA) commonly occurs as intermediates in DNA metabolic pathways. The ssDNA binding protein, RPA, not only protects the integrity of ssDNA, but also directs the downstream factor that signals or repairs the ssDNA intermediate. However, it remains unclear how these enzymes/factors outcompete RPA to access ssDNA. Using the budding yeast *Saccharomyces cerevisiae* as a model system, we find that Dna2 — a key nuclease in DNA replication and repair — employs a bimodal interface to act with RPA both in cis and in trans. The cis-activity makes RPA a processive unit for Dna2-catalyzed ssDNA digestion, where RPA delivers its bound ssDNA to Dna2. On the other hand, activity in trans is mediated by an acidic patch on Dna2, which enables it to function with a sub-optimal amount of RPA, or to overcome DNA secondary structures. The trans-activity mode is not required for cell viability, but is necessary for effective double strand break (DSB) repair.

[1] Department of Molecular and Cellular Biochemistry, Indiana University, Bloomington, IN 47405, USA. [2] Center for Quantitative Biology, Peking-Tsinghua Center for Life Sciences, Academy for Advanced Interdisciplinary Studies, Peking University, Beijing 100871, China. [3] Department of Molecular and Human Genetics, Baylor College of Medicine, Houston, TX 77030, USA. [4] Department of Chemistry, Biological Mass Spectrometry Facility, Indiana University, Bloomington, IN 47405, USA. [5] These authors contributed equally: Jiangchuan Shen, Yiling Zhao and Nhung Tuyet Pham. ✉email: zhiqi7@pku.edu.cn; hniu@indiana.edu

Single-stranded DNA (ssDNA) is a common intermediate of DNA replication and repair. During DNA replication, ssDNA is transiently exposed as the template for lagging strand synthesis and comprises the 5′-flap DNA that must be removed during lagging strand maturation[1]. Lesion removal in mismatch repair, nucleotide excision repair, and long-patch base-excision repair also generates ssDNA intermediates subject to gap-filling repair[2]. During homologous recombination (HR), nucleolytic processing of double-strand breaks (DSBs) produces 3′-ssDNA that enables Rad51-catalyzed homology search and strand exchange[3].

In eukaryotes, replication protein A (RPA) readily occupies ssDNA, protects it from nucleolytic attack, and directs enzymes involved in lesion repair. RPA-coated ssDNA also marks DNA-damage sites and triggers ATR/Mec1-dependent DNA replication and DNA-damage checkpoints[4]. Consequently, a failure to remove RPA-coated ssDNA such as the 5′-flap formed during lagging-strand maturation, may lead to a prolonged cell cycle arrest and cell death[5]. Removal of RPA-coated ssDNA intermediates can be accomplished by DNA polymerase-catalyzed duplex generation, or replacement with downstream repair factors such as the Rad51 recombinase. In addition, the evolutionarily conserved Dna2 endonuclease provides a unique means to remove RPA-coated ssDNA from an open 5′-end. During lagging strand synthesis, Dna2 removes RPA-coated long 5′-flap and allows Fen1 nuclease to further complete lagging strand maturation[1]. This function of Dna2 is essential for viability in yeast[6]. The lethality of *dna2* mutants, however, can be rescued by the inactivation of the DNA-replication checkpoint[5] or Pif1[7], a 5′-to-3′ DNA helicase responsible for the formation of long 5′-flaps during lagging-strand synthesis[8]. Besides DNA replication, Dna2 also plays a pivotal role in DSB repair, where it digests the RPA-coated 5′-strand unwound by the Sgs1 helicase to aid DSB end processing[9–12]. The Sgs1–Dna2 pair serves as an alternative means in addition to the Exo1 nuclease to catalyze the long-range resection of DSB following its initial trimming by the action of Mre11–Rad50–Xrs2 (MRX) and Sae2[9,10]. Regarding the interplay with RPA, a central question concerns how the downstream enzymes/factors outcompete RPA to access ssDNA during DNA replication and repair.

Here, we combine biochemical, single-molecule, and genetic approaches to provide evidence that RPA, rather than acting passively[13], employs a gating mechanism to function as an active unit for the Dna2 nuclease. Within this framework, RPA allows its bound ssDNA to be released to Dna2 within the ternary complex formed by Dna2, RPA, and 5′-ssDNA. Our study also reveals a trans-acting element supported by a bimodal interface between Dna2 and RPA, which allows Dna2 within the Dna2–RPA–ssDNA ensemble to interact with other RPA molecules and maintains its full activity when RPA becomes limiting or DNA secondary structures are encountered. We also report that a separation-of-function mutant of *DNA2*—*dna2-AC*—which inactivates the trans-acting element, fully supports cell viability but causes a major defect in DSB repair, presumably due to a failure to resolve a critical recombination intermediate.

## Results

### RPA promotes long ssDNA digestion by Dna2 in addition to its role in Dna2 recruitment.
Extensive ssDNA digestion by Dna2 nuclease likely occurs during double-strand break processing and lagging-strand maturation[9,10,14]. The processivity of Dna2 nuclease, however, has not been determined. In biochemical assays, Dna2 nuclease activity is salt sensitive. Increasing the KCl concentration from 50 to 250 mM almost completely inactivates Dna2 on the initial cleavage of a 5′-overhanging DNA (Supplementary

Fig. 1a–c). The presence of RPA has a minimal effect on 5′-incision by Dna2 under low-salt conditions (50 mM KCl) (Supplementary Fig. 1a), but becomes stimulatory under physiological salt conditions (150 mM KCl) (Supplementary Fig. 1b) and indispensable at 250 mM KCl (Supplementary Fig. 1c). Removal of amino acid 1–180 of the Rfa1 subunit (RPA–ΔN), which disrupts the reported interface between Dna2 and RPA but does not affect DNA binding by RPA (Supplementary Fig. 1d)[15], drastically abrogated Dna2 stimulation by RPA (Supplementary Fig. 1e)[16]. Thus, RPA recruits Dna2 to the 5′-end of ssDNA as reported[1], which can largely be offset in vitro by reducing the salt in reactions. When Dna2 was challenged with a long ssDNA substrate (562 bases), the initial 5′-cleavage was comparable to the short substrate under 50 mM KCl (Fig. 1a), which, again, was not affected by the addition of RPA (Fig. 1a). However, Dna2-catalyzed digestion of the same 562-nt ssDNA substrate that was randomly labeled was reduced by nearly 3-fold, which can be fully rescued by the presence of RPA, but not RPA–ΔN (Fig. 1b). Knowing that Dna2 is proficient in cleaving the 5′-ssDNA end under 50 mM KCl, we hypothesized that in addition to recruitment of Dna2 to ssDNA, RPA also renders Dna2-catalyzed ssDNA digestion processive in a manner that requires the Rfa1-N domain.

### Processive digestion of RPA-coated ssDNA by Dna2 in single-molecule DNA Curtains.
To directly test the premise that RPA renders Dna2-catalyzed ssDNA digestion processive, we utilized the high-throughput single-molecule imaging technology, DNA Curtains[17,18], to visualize the behavior of quantum-dot (QD)-labeled Dna2 on individual RPA–GFP-coated ssDNA molecules with a free 5′-end (Fig. 1c, d & Supplementary Movie 1, Supplementary Fig. 2a–g). In the presence of 0.1 nM free RPA–GFP and 50 mM NaCl in the buffer, both Dna2 and dna2–D657A, a nuclease-dead version of Dna2[19], were recruited and stably bound to the 5′-terminus of ssDNA with a lifetime of 480 ± 89 s (mean ± s.e.m., $N = 33$) and 528 ± 110 s (mean ± s.e.m., $N = 47$), respectively (Fig. 1e). However, only wild-type Dna2 displayed a measurable movement along DNA accompanied by the shortening of RPA-coated ssDNA presumably emanating from ssDNA digestion (Fig. 1f and h). Fitting the histogram of processivity of Dna2 traces with three Gaussian functions revealed three peaks. Peak 1 represents Dna2 molecules with minimal measurable movement (42 ± 91 (nm)). Peaks 2 and 3 represent Dna2 molecules with a processivity of 369 ± 80 nm and 1058 ± 86 nm, respectively (Fig. 1f). Notably, the majority of Dna2 molecules remain bound to DNA for 440 ± 107 s (mean ± s.e.m., $N = 9$) following the observable DNA digestion (Fig. 1e). Increasing the NaCl concentration to 150 mM, neither drastically reduced the lifetime (414 ± 44 s, mean ± s.e.m., $N = 47$) (Fig. 1e), nor changed the processivity of Dna2 on RPA-coated ssDNA (Fig. 1g). However, after removing RPA from the reaction buffer, the processivity of Dna2 (Fig. 1i) was greatly reduced and could not be distinguished from dna2–D657A (Fig. 1h). Based on a rough conversion of approximately 0.27 nm per nucleotide (nt) (Methods), Dna2 molecules in peaks 2 and 3 with 50 mM NaCl are capable of processively digesting 1367 ± 296 nt and 3919 ± 319 nt of RPA-coated ssDNA per single turnover reaction. The presence of ATP (1 mM) in the system had little impact on the processivity of Dna2 (Supplementary Fig. 2h–j), which suggests that the motor activity of Dna2 contributes minimally to the processivity of Dna2-catalyzed digestion of RPA-coated ssDNA.

### RPA promotes Dna2-catalyzed long ssDNA digestion by forming a stable ternary complex at the 5′-end of ssDNA.
In DNA Curtains, replacing RPA–GFP with RPA–ΔN–GFP failed to recruit Dna2 to the 5′-end of ssDNA even under 50 mM NaCl.

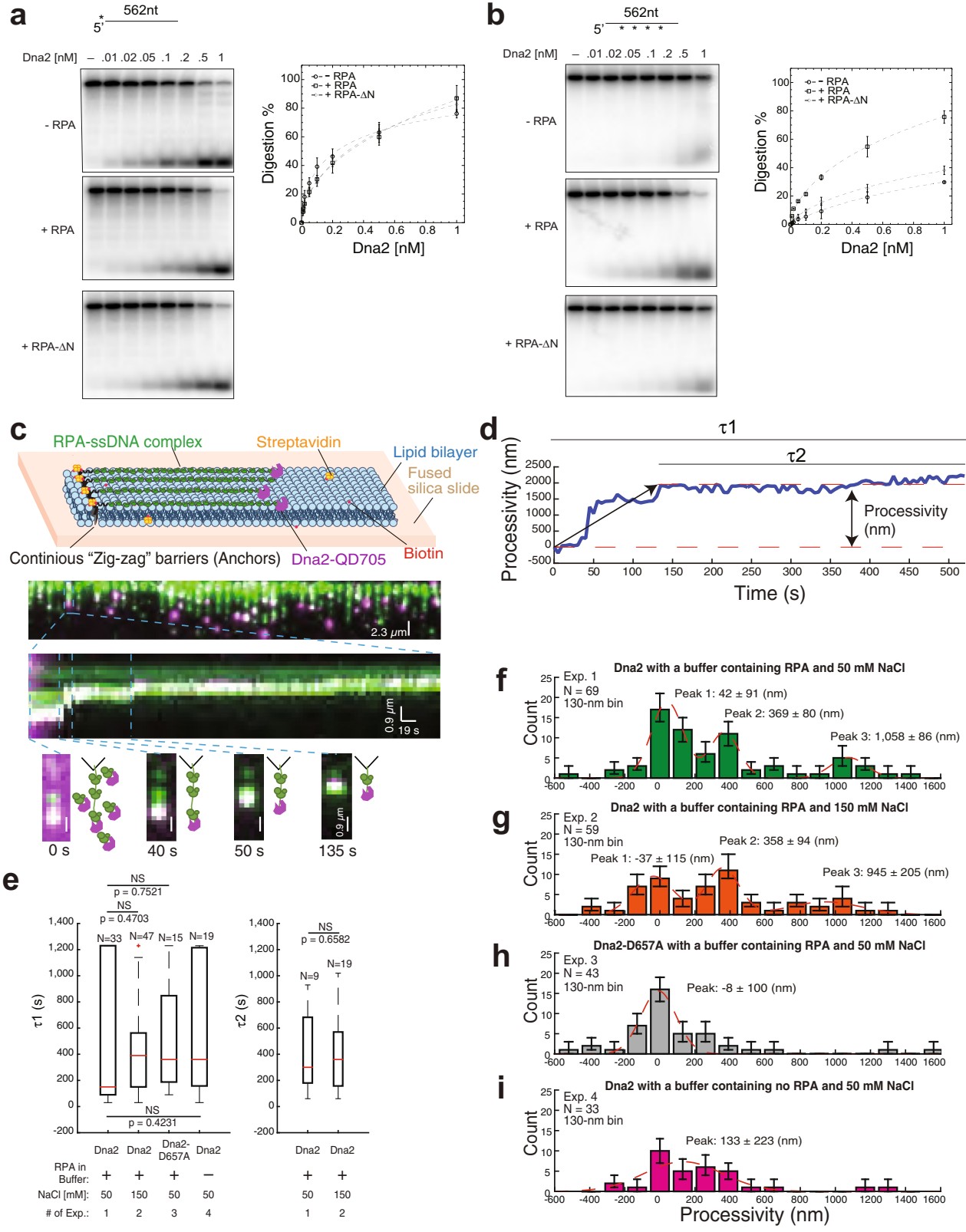

However, to our surprise, injection of both Dna2 and RPA–GFP into RPA–ΔN–GFP-coated ssDNA Curtains enables Dna2 to digest RPA–ΔN–GFP-coated ssDNA with a processivity of 516 ± 80 nm (Fig. 2a–d), which indicates that Dna2 turnover from one RPA to another may not be required in order to processively digest ssDNA. In biochemical assays with 5 nM randomly labeled 562-nt ssDNA substrate, a 2-fold Dna2 stimulation was observed with an equal molar amount of RPA (5 nM) (Supplementary Fig. 3a) and was not affected by precoating ssDNA with RPA–ΔN (Fig. 2e). The stimulation is nearly saturated with 20 nM RPA, corresponding to around four RPA molecules per ssDNA molecule, which is nearly five times less than the amount needed to fully coat the 562-nt ssDNA substrate (Supplementary Fig. 3a). Collectively, these results led us to surmise that the Dna2–RPA ensemble may be able to act

**Fig. 1 RPA renders Dna2-catalyzed ssDNA digestion processive. a, b** Digestion by titrated Dna2 on 5′-labeled (**a**) and internally labeled (**b**) 562-nt ssDNA without RPA, with RPA (200 nM), or RPA–ΔN (200 nM) under 50 mM KCl. For quantification, mean values ±s.d. from three independent experiments were plotted. **c** Schematic of DNA Curtains experimental design, wide-field image of Dna2 (magenta puncta) digesting ssDNA coated by RPA (green), and representative kymograph showing Dna2 digestion. **d** The trajectory of Dna2 molecule positions versus real-time in **c** was plotted. The processivity was defined as the distance of Dna2 movement. $\tau1$ was defined as the dwell time of Dna2 binding to RPA–ssDNA complex, and $\tau2$ was defined as the dwell time of Dna2 on the RPA–ssDNA complex after digestion. **e** Boxplot of $\tau1$ and $\tau2$ for different experimental conditions (Supplementary Table 1). The total number of Dna2 digestion events in different experimental conditions examined over more than three DNA Curtains experiments ($n \geq 3$) was indicated in all boxplots. For the boxplot, the red bar represents the median. The minima of the box represents $25^{th}$ percentiles, and the maxima is $75^{th}$ percentiles. Most extreme data points are covered by the whiskers except outliers. The '+' symbol is used to represent the outliers. Statistical significance was analyzed using the unpaired t-test for two groups. *p*-value: two-tailed; *p*-value style: GP: 0.1234 (ns), 0.0332 (*), 0.0021 (**), 0.0002 (***), <0.0001 (****). Confidence level: 95%. **f-i** Processivity distribution of single Dna2 digestion for different experimental conditions (Supplementary Table 1): 130–nm bin. The total number of Dna2 digestion events (*N*) was as labeled. Each experimental condition was examined over more than three DNA Curtains experiments ($n \geq 3$). Error bars were obtained through the bootstrap analysis. For any normally distributed dataset, 68.27% of the values lie within one standard deviation of the mean, therefore, our choice of 70% confidence intervals for the bootstrapped data provides a close approximation to expectations for one standard deviation from the mean. The data were fitted with Gaussian functions (red dash line). The errors represented 95% confidence intervals obtained through Gaussian function fitting. Source data are provided as a Source Data file.

as a stable unit to processively digest ssDNA. In agreement with this hypothesis, RPA and dna2–D657A form a ternary complex on 5′-overhanging DNA, similar to what is reported (Fig. 2f)[1]. Replacing the 5′-overhanging DNA with 3′-overhanging DNA or a gapped duplex reduced the amount of ternary-complex formation 3–6 fold (Fig. 2f). Addition of BS3 (bis[sulfosuccinimidyl]), a cross-linker of 11.4 Å in length that has an amine-reactive *N*-hydroxysulfosuccinimide (NHS) ester at both ends, to the mixture of Dna2 (1 μM), RPA (1 μM), and $dT_{(30)}$ oligonucleotides, led to the formation of a new species on SDS-PAGE (Fig. 3a). The resulting cross-linked product with a molecular weight corresponding to a Dna2 monomer plus an RPA heterotrimer was minimally detected in the absence of oligonucleotides (Fig. 3a) or the presence of oligonucleotides of 15 bases in length or shorter (Supplementary Fig. 3b).

**The N-terminus of Rfa1 and the N-OB-fold region in Dna2 interact to support their ternary-complex formation with ssDNA.** Bimodal interaction between Dna2 and RPA has been proposed[16]. To probe the interface between Dna2 and RPA in the context of their ternary complex with ssDNA, we analyzed the cross-linked product of Dna2 and RPA in the presence of $dT_{(30)}$ oligonucleotides by mass spectrometry. Notably, the OB-fold region in Dna2 that bridges its N-terminus and the nuclease domain was cross-linked to the linker region that bridges the N-terminus of Rfa1 and the DNA-binding domain A (DBD-A) (Fig. 3b and Supplementary Table 2). Additionally, extensive contact between the N-terminus of Dna2 and the N-terminus of Rfa1 was also detected (Fig. 3b and Supplementary Table 2). Consistent with these findings, dna2–Δ501 N that lacks the N-terminus of Dna2 and its adjoining OB fold failed to form a ternary complex with Rfa1–NAB on $dT_{(20)}$ ssDNA (Supplementary Fig. 3c). In addition, dna2–Δ501 N is severely defective in nuclease activity and failed to be stimulated by RPA (Supplementary Fig. 3d), which indicates a direct role of the Dna2 OB fold in nucleolytic digestion. Coincidentally, the *dna2-1* temperature-sensitive mutant[19], dna2-P504S, resembles dna2–Δ501 N and is defective not only in nuclease activities (Supplementary Fig. 3e) but also for ternary-complex formation (Supplementary Fig. 3f).

**Rfa1-N-DBD-A/B polypeptide is proficient in supporting Dna2-catalyzed ssDNA digestion.** RPA occupies ssDNA with a polarity where the N-terminus and the DBD-A domain of Rfa1 lay toward the 5′-end of ssDNA[20,21]. The fact that the interface between Dna2 and RPA was mapped to the N-terminus of Rfa1 prompted us to isolate the minimal portion of RPA that supports

ternary complex formation with Dna2. Neither Rfa1-N (amino acid 1–180) nor the Rfa1-N-DBD-A (amino acid 1-299) were able to form a ternary complex with Dna2 (Fig. 3c). However, Rfa1-N-DBD-A/B (Rfa1–NAB, amino acid 1–435), a polypeptide that contains the N, DBD-A, and DBD-B domains from Rfa1 and binds ssDNA ~100 times weaker than wild-type RPA[22], is fully capable of ternary-complex formation with Dna2 (Fig. 3c) and stimulates Dna2-catalyzed digestion of both short and long ssDNA substantially (3- and 2-fold, respectively) (Supplementary Figs. 4a and 3d). We thus conclude that the polypeptide of Rfa1–NAB represents the major portion of RPA that synergizes with Dna2. A DNA-binding mutant of either DBD-A domain (Rfa1–NA⁻B) or DBD-B domain (Rfa1–NAB⁻)[23] failed to stimulate Dna2 ssDNA cleavage (Fig. 3e), but a fusion of the N-terminus of yeast Rfa1 (amino acid 1–180) to the DBD-A/B polypeptide from human RPA (amino acid 181–435) partially activated human DBD-A/B for the stimulation of yeast Dna2 (Fig. 3e). Thus, the N-domain in Rfa1 plays a major role in its synergy with Dna2, while the major contribution of DBD-A and DBD-B domains is likely to bind substrate DNA.

**Fusion of Rfa1–NAB to Dna2–Δ248 N activates Dna2 for long ssDNA digestion in the absence of RPA, but fails to overcome DNA structures.** Knowing that Rfa1–NAB is the minimal portion of RPA capable of synergizing with Dna2, we fused Rfa1–NAB to the N-terminus of dna2–Δ248 N, an N-terminal truncation of Dna2 lacking the CDK phosphorylation sites but fully functional in vitro[24]. The fusion protein referred to as NAB–dna2–Δ248 N hereafter forces Rfa1–NAB and dna2–Δ248 N to function as a single unit. On 5′-overhanging DNA with a 40-nt ssDNA tail, NAB–dna2–Δ248 N displayed activities only marginally less compared with a mixture of free Dna2 and RPA both on the initial 5′-cleavage and on the progressive digestion toward the 3′-end (Figs. 4a and b), which supports the conclusion that Dna2 and RPA function as a stable unit in ssDNA digestion. However, to our surprise, when NAB–dna2–Δ248 N was challenged with the 562-nt ssDNA, NAB-dna2-Δ248 N was defective not only in the digestion of internally labeled substrate compared with free Dna2 and RPA (Supplementary Fig. 4b) but also in the initial 5′-cleavage (Supplementary Fig. 4c). Unlike Dna2, reducing the KCl concentration to 50 mM did not enhance the 5′ cleavage by NAB–dna2–Δ248 N on 562-nt ssDNA (Supplementary Fig. 4c). To probe the defect of NAB–dna2–Δ248 N on long ssDNA, we generated a 5′-overhanging DNA with a 140-nt ssDNA region adjoined with a 20-bp duplex based on the sequence at the 5′-end of the 562-nt ssDNA. To our surprise, accumulation of pausing

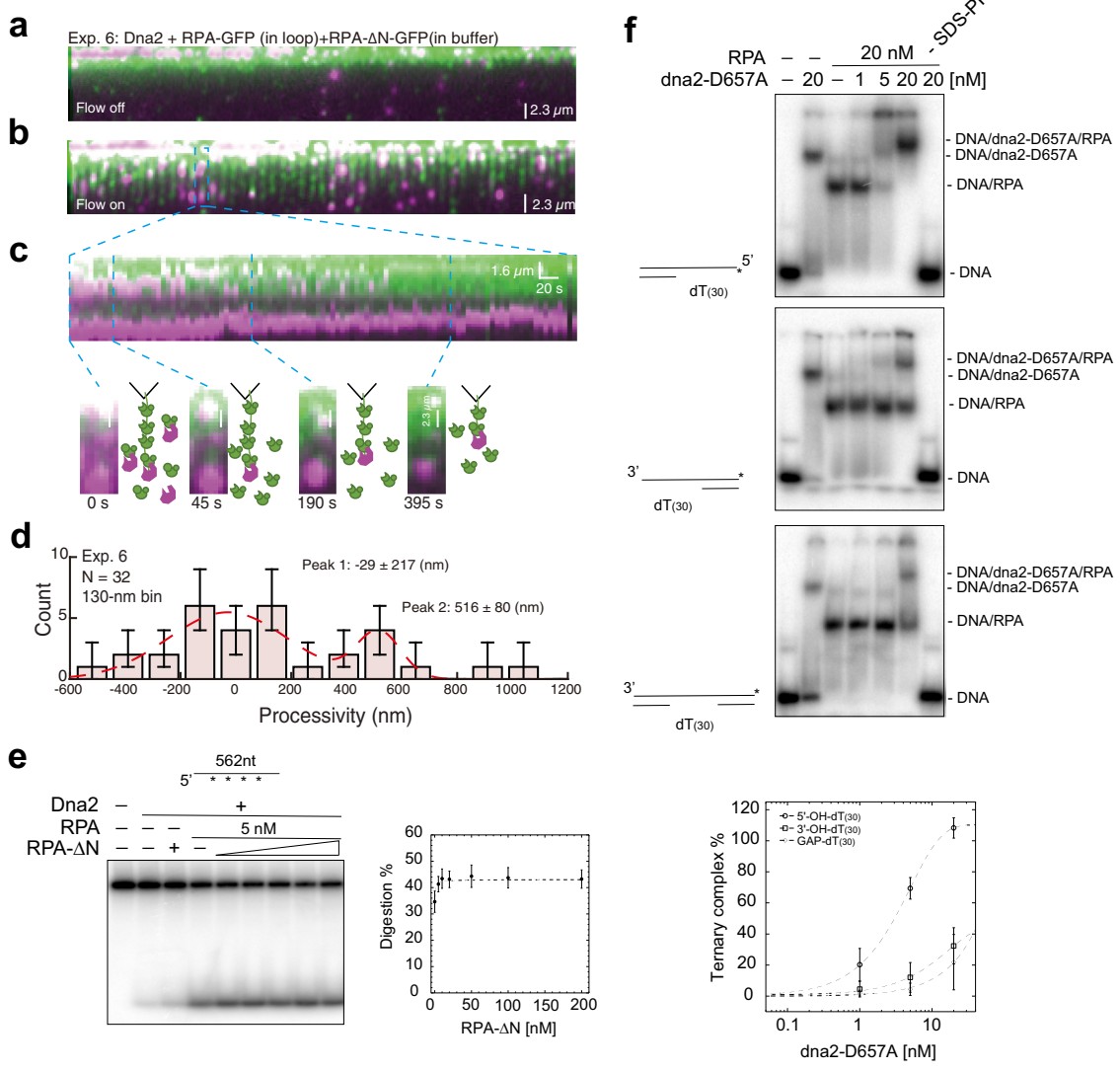

**Fig. 2 Dna2–RPA ensemble catalyzes ssDNA digestion by forming a stable unit at the 5′-end of ssDNA. a–d** Wide-field image of Dna2 (magenta punctum)–RPA–ΔN–GFP complexes digesting the ssDNA–RPA–ΔN–GFP complexes (green) (Exp. 6 in Supplementary Table 1). **a** Flow off; (**b**) flow on; (**c**) representative kymograph showing the process of digestion. Snapshots at time 0, 45, 190, and 395 seconds were shown. **d** Processivity distribution, 130-nm bin. $N = 32$, which was the total digestion events examined over three DNA Curtains experiments ($n = 3$). Error bars were obtained through the bootstrap analysis. For any normally distributed dataset, 68.27% of the values lie within one standard deviation of the mean, therefore, our choice of 70% confidence intervals for the bootstrapped data provides a close approximation to expectations for one standard deviation from the mean. The data were fitted with Gaussian functions (red dash line). The errors represented 95% confidence intervals obtained through Gaussian function fitting. **e** Digestion by Dna2 on internally labeled 562-nt ssDNA with stoichiometry amount of RPA in the presence of titrated amount of RPA–ΔN under 150 mM KCl. For quantification, mean values ±s.d. from three independent experiments were plotted. **f** The dna2-D657A mutant preferentially forms a ternary complex with RPA on 5′-dT$_{(30)}$-overhanging ssDNA (5′-OH-dT$_{(30)}$), in comparison with 3′-dT$_{(30)}$-overhanging ssDNA (3′-OH-dT$_{(30)}$) and dT$_{(30)}$-gap ssDNA (GAP- dT$_{(30)}$) under 150 mM KCl. For quantification, mean values ±s.d. from three independent experiments were plotted. Source data are provided as a Source Data file.

products 70-80 bases from the 5′-end was observed in the reactions with NAB–dna2–Δ248 N, but not with free Dna2 and RPA or NAB (Fig. 4c). In this region, sequence analysis predicted the formation of a stable hairpin. Removal of the hairpin eliminated the pausing products (Fig. 4d), which suggests the hairpin acts as a structural barrier for NAB–dna2–Δ248 N. Importantly, the addition of NAB polypeptide had no impact on the digestion by NAB–dna2–Δ248 N (Fig. 4c), which strongly suggests that a trans action between Dna2 and RPA may aid in overcoming the structural barrier. On the substrate without a hairpin, the overall reaction rate was not affected (Fig. 4d), likely due to a turnover problem at the junction of single/double stranded DNA. Extending the duplex region to 40 bp had no effect on the reaction rate (Fig. 4d), which confirms that the

failure of turnover is not DNA sequence-dependent. Again, the reaction rate was not affected by the addition of the NAB poly-peptide (Fig. 4d). An ssDNA length titration revealed a jump in the digestion rate by NAB–dna2–Δ248 N when the ssDNA length dropped from 80 nt to 60 nt (Supplementary Fig. 4d). Hence, the trans action between Dna2 and RPA is likely needed for Dna2 turnover following long ssDNA digestion.

**Rfa1–Y193A mutant partially supports Dna2 recruitment but is defective in supporting processive ssDNA digestion by Dna2.** In our model, to allow Dna2 and RPA to function as a stable unit, a gate likely exists in RPA to enable the delivery of ssDNA from RPA to Dna2. Knowing that aromatic residues play an important

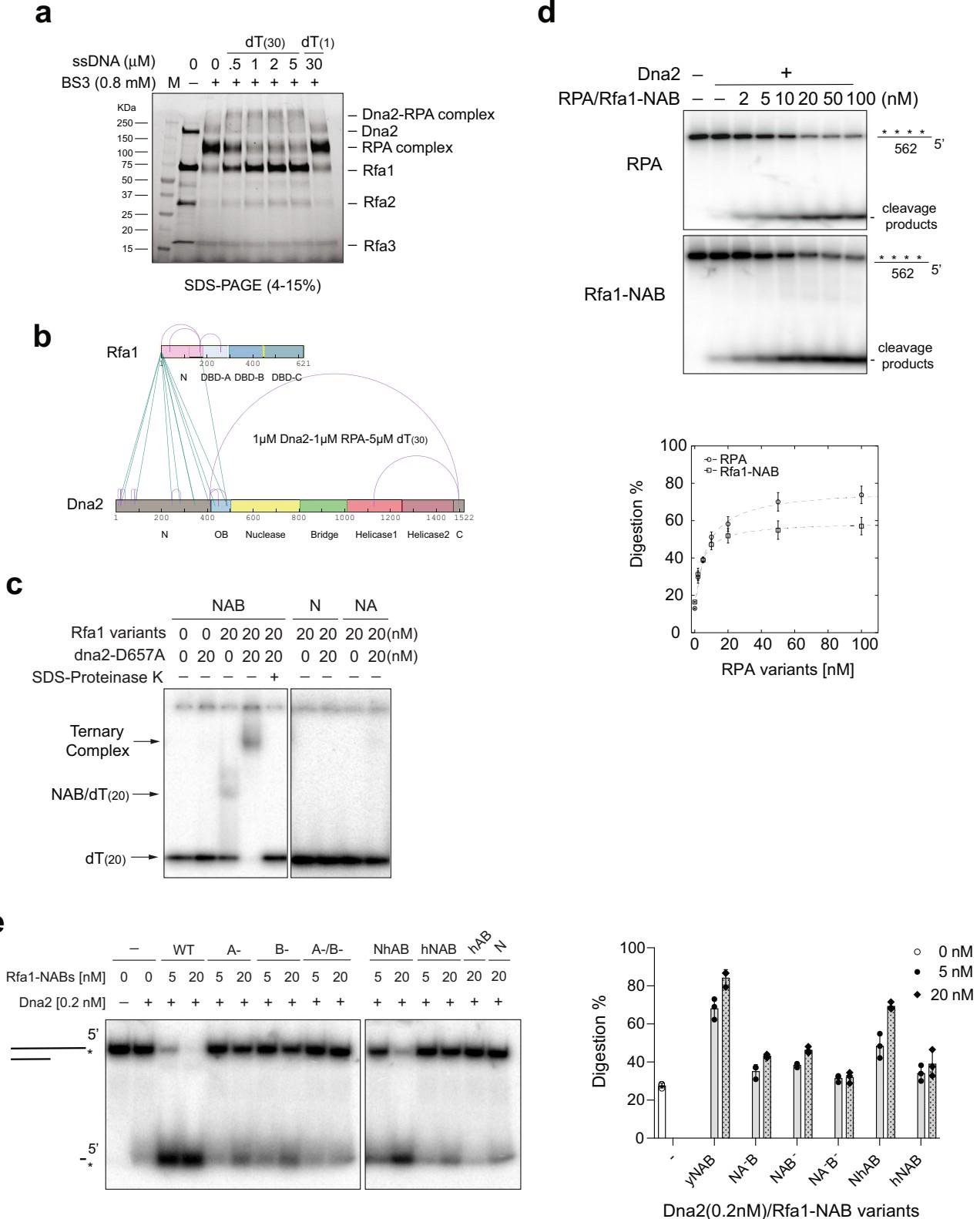

role in ssDNA binding by RPA, we conducted an alanine-scanning of the fifteen aromatic amino acid residues from yeast Rfa1-N (1–196) (Supplementary Fig. 5a). The total fourteen rfa1–NAB mutants (F11A, F15A, Y21A, Y29A, Y32A, Y55A, F69A, Y96A, F103A, F119A, Y123A/F124A, F171A, F185A, and Y193A) with each of single or two adjacent aromatic amino acids from Rfa1-N changed into alanines respectively and revealed that

six of the rfa1–NAB mutants, F15A, Y29A, Y55A, Y96A, Y123A/F124A, and Y193A, were partially defective in stimulating Dna2's initial cleavage of a 5′-overhanging DNA (Fig. 5a). Among those six mutants, rfa1–NAB–Y193A, in particular, is defective not only in supporting the initial 5′-cleavage by Dna2 but also in supporting Dna2's processive cleavage of the same 5′-overhanging DNA (Fig. 5b, c and Supplementary Fig. 5b). In DNA Curtains,

**Fig. 3 The Rfa1–NAB polypeptide functions as the core domain from RPA to support Dna2-catalyzed ssDNA digestion. a** Dna2 formed cross-linked species with RPA in the presence of $dT_{(30)}$ ssDNA (0–5 µM) as resolved on a 4–15% SDS-PAGE. The experiments were carried out once. **b** The diagram of the mass-spectrometry-captured $BS^3$-containing dipeptides (see Supplementary Table 2 for detailed cross-linked dipeptides) with intermolecular species shown in cyan lines and intramolecular species in red curves. **c** Ternary-complex formation was observed between dna2–D657A (20 nM), $dT_{(20)}$ ssDNA (5 nM) and Rfa1–NAB (20 nM), but not Rfa1-N or Rfa1-NA. The experiments were repeated three times. **d** Comparison of titrated RPA and Rfa1–NAB (0–100 nM), on the digestion of internally labeled 562-nt ssDNA (5 nM) by Dna2 (1 nM). For quantification, mean values ± s.d. from three independent experiments were plotted. **e** Digestion by Dna2 (0.2 nM) on 5′-labeled 40-nt 5′-overhanging ssDNA (5 nM) with indicated amount of Rfa1–NAB and hRPA70–NAB variants. For quantification, mean values ± s.d. from three independent experiments were plotted. Source data are provided as a Source Data file.

RPA complex harboring rfa1–Y193A mutant, while allowing 80% of recruitment of Dna2 molecules comparing with wild-type RPA (Fig. 5d–f), resulted in minimal measurable Dna2 movement ($-121 \pm 208$ (nm)) (Fig. 5e). Thus, the linker region between Rfa1-N and Rfa1–DBD-A may play an important role in the gating of ssDNA.

**dna2-AC mutant is defective in the interaction with RPA in solution, but proficient in ternary-complex formation with RPA and ssDNA.** Besides Dna2, Rfa1-N recruits multiple proteins in DNA replication and repair. Most of such interactions are mediated by an acidic patch from the partner protein and the basic cleft in Rfa1-N. *rfa1-K45E* mutant, initially identified as *rfa1-t11*, contains a charge-reversal mutation of a key lysine residue in the basic cleft and is defective in DNA repair[25]. We created and tested rfa1–t11(rfa1–K45E) mutant, which showed a 4-fold reduction in the stimulation of Dna2's initial cleavage of the 5′-overhanging DNA (Fig. 5g) and is partially defective in the ternary-complex formation with Dna2 and ssDNA (Fig. 5h). Given the pleiotropic phenotype of *rfa1-K45E* mutant, we decided to search for the corresponding *DNA2* mutants for phenotypic dissection. Aligning the N-terminus of Dna2 from multiple yeast species revealed two conserved acidic patches located at amino acid 210–219 and amino acid 355–366 (Fig. 6a). The first acidic patch has been reported to bridge Dna2 interaction with Ctf4[26] and purified dna2–Δ248 N interacts with RPA as well as the full-length Dna2[24]. Thus, we focused on the second acidic patch and created a compound mutant (dna2–D357A,E358A,D361A,D362A, and E366A) with all acidic residues in this motif mutated into alanines. The resulting mutant, namely dna2-AC, is defective in pull-down assays against RPA (Fig. 6b), but surprisingly can still form a stable ternary complex with RPA and $dT_{30}$ (Fig. 6c). In the microscale thermophoresis analysis, a biophysical technique to characterize biomolecule interactions based on the temperature-induced change in fluorescence of the target, the dissociation constant of dna2-AC and RPA was $805 \pm 22$ nM, which is nearly 4-fold higher than the dissociation constant of Dna2 and RPA ($185 \pm 7$ nM) (Fig. 6d), which affirms the defect of dna2-AC in RPA interaction.

**dna2-AC mutant cells are viable, but defective in double-strand break repair.** To query the importance of the interface between the Dna2 AC motif and Rfa1-N in cells, we generated a *dna2-AC* mutant strain in *Saccharomyces cerevisiae*. Unlike *dna2Δ*, *dna2-AC* is viable with no apparent growth defect and is not synthetically lethal with *exo1Δ* (Fig. 6e). In a physical assay to monitor resection of a DSB generated by HO endonuclease (Fig. 6f)[27], consistent with the role of Dna2 in long-range resection[10], the *dna2-AC* mutant was similar to wild type in resection at the vicinity of the break but showed defects in resection at 10 kb away from the HO break, which is synthetic with *exo1Δ* (Fig. 6g)[10,24]. Biochemically, we also observed a 2–3-fold reduction in resection by the Sgs1–Dna2 machinery when Dna2 was substituted by dna2-AC (Supplementary Fig. 6a–b). Interestingly, the *dna2-AC*

mutant is hypersensitive to phleomycin, a DNA-damaging agent known to cause double-strand breaks in cells, but not hydroxyurea (HU) or camptothecin (CPT), two drugs that cause single-strand breaks by either stalling replication forks or trapping topoisomerase I (Fig. 6e). However, the phleomycin sensitivity of *dna2-AC* is not synthetic with *exo1Δ*, but suppressed by the concurrent deletion of *PIF1* (Fig. 6e). Thus, the sensitivity of the *dna2-AC* mutant to phleomycin can not be simply explained by its defect in DNA end resection and is likely related to the additional role of Dna2 in DSB repair, where it processes Pif1-generated substrates.

**dna2-AC-catalyzed ssDNA digestion is less processive and sensitive to DNA structure.** We further characterized dna2-AC nuclease activities in our biochemical and single-molecule assays. In biochemical assays, dna2-AC is partially defective in its stimulation by RPA (~2-fold) (Supplementary Fig. 7a) on the initial 5′-cleavage, despite the defect being apparent only when RPA was limiting. However, on long ssDNA (562 nt), a consistent 3-fold reduction was observed with dna2-AC under various concentrations of RPA (Fig. 7a). Interestingly, on the 140-nt 5′-overhanging DNA with the hairpin structure, dna2-AC also displayed sensitivity to the hairpin structure in addition to a defect in initiation, especially when the concentration of RPA was suboptimal (Fig. 7b) which indicates a role of the AC domain in the in-trans action between Dna2 and RPA to overcome the barrier of DNA structures. Consistent with this model, NAB–dna2–Δ405 N, which lacks the AC domain, displayed nuclease activities that are comparable to NAB–dna2–Δ248 N (Supplementary Fig. 7b–d). In DNA Curtains, the recruitment of dna2-AC is about 2-fold less compared with wild type (Fig. 7c, d). The recruited dna2-AC had an average lifetime of $692 \pm 107$ s (mean ± s.e.m., N = 17) (Fig. 7e), which is similar to wild type (Fig. 1e) and consistent with the observed stable ternary-complex formation of dna2-AC and RPA on ssDNA (Fig. 6c). However, the processivity of dna2-AC decreased to $377 \pm 112$ nm under 50 mM NaCl (Fig. 7f) and $131 \pm 127$ nm under 150 mM NaCl (Fig. 7g). Further removal of RPA from the 50 mM NaCl buffer decreased the processivity of dna2-AC (Fig. 7h) to the undetectable level but did not affect Dna2 recruitment (Fig. 7d). We thus surmise that the Dna2 AC domain may contribute to both the in-cis- and in-trans-interactions with RPA. The in-cis interaction may aid Dna2 recruitment, while the in-trans-interaction following Dna2 recruitment is critical for Dna2 to overcome structural barriers and to achieve the maximum level of processivity that is required for successful DSB repair.

## Discussion

To summarize, our results suggest that Dna2 becomes a processive enzyme only after ensembled with RPA (Fig. 8), which is most active on ssDNA shorter than 60 nt. For longer ssDNA with secondary structures, the Dna2 AC domain from the Dna2–RPA ensemble may need to act in-trans with other RPA

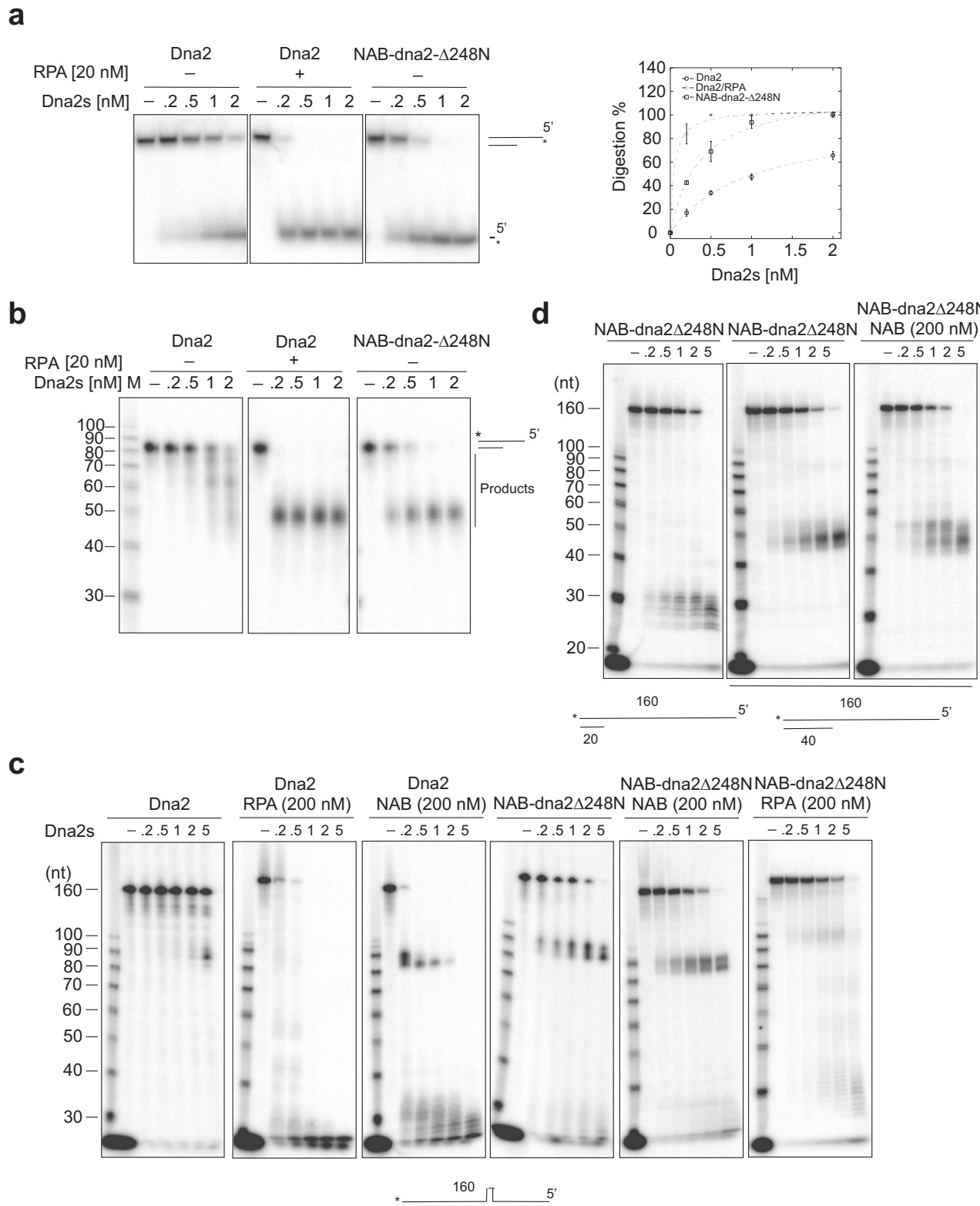

molecules to overcome DNA secondary structures. Notably, the NAB–dna2–Δ248 N fusion can digest ssDNA coated with RPA (Fig. 4c), but it fails to turnover from a long ssDNA overhang (Fig. 4d). Thus, the cis-action in the Dna2–RPA ensemble allows the removal of RPA from ssDNA likely through DNA degradation, while the trans-action may be required to facilitate the dissembling of the ternary complex and the turnover of Dna2

following overhang digestion. Our model further suggests that RPA entails a gating mechanism that can actively deliver ssDNA to Dna2. Considering that the yeast Rfa1-N can partially activate the DBD-A/B from human RPA, our model predicts that the gate for the DNA transportation likely resides at the N-terminus of RPA. Mutation analysis allowed us to map the gate to a conserved region immediately upstream of the DBD-A domain in

**Fig. 4 The fusion of Rfa1–NAB domain to dna2Δ248 N supports processive ssDNA digestion on nonstructured ssDNA. a** Digestion by titrated Dna2 (0–2 nM) or NAB–dna2Δ248 N (0–2 nM) on 5′-labeled 40-nt 5′-overhanging ssDNA (5 nM) without or with RPA (20 nM). For quantification, the mean values ± s.d. from three independent experiments were plotted. **b** Digestion by titrated Dna2 (0 nM to 2 nM) or NAB–dna2Δ248 N (0 nM to 2 nM) on 3′-labeled 40-nt 5′-overhanging ssDNA (5 nM) without or with RPA (20 nM). **c** Digestion of 3′-labeled 140-nt 5′-overhanging ssDNA (5 nM) (hairpin-containing) by titrated Dna2 (0–5 nM) without RPA, with RPA (200 nM) or Rfa1–NAB (200 nM), and by titrated NAB–dna2Δ248 N (0–5 nM) without or with Rfa1–NAB (200 nM) or RPA (200 nM). **d** Digestion of 3′-labeled 140-nt 5′-overhanging ssDNA (5 nM) (hairpin-removed) by titrated NAB–dna2Δ248 N (0–5 nM) and digestion of 3′-labeled 120-nt 5′-overhanging ssDNA (5 nM) (hairpin-removed) by titrated NAB–dna2Δ248 N (0–5 nM) without or with Rfa1–NAB (200 nM). The experiments in b–d were repeated three times. Source data are provided as a Source Data file.

RPA. Our study thus provides a new paradigm on recognizing the dynamic nature of RPA, where, through different modes of interaction, RPA can actively deliver its bound ssDNA to downstream enzymes/factors and coordinate with them to further resolve DNA secondary structures encountered. Dna2 physically interacts with Sgs1 during DNA end resection in yeast[11,12,28] and is regulated by CtIP in human cells[29,30]. Although the reported stimulation of human DNA2 by CtIP is ATP-dependent and likely relies on the motor activity of human DNA2[29], the Sgs1 helicase, which also interacts with Dna2, on the other hand, possesses an acid patch that has been shown to interact with Rfa1-N[31]. Given the resection defects that we observed with dna2-AC (Fig. 6g and Supplementary Fig. 6), it will be interesting to further investigate the interplay of AC motifs from Sgs1 and Dna2 with Rfa1-N during DNA end resection. Besides Dna2, numerous other protein factors also function with RPA in DNA replication and repair, including DNA polymerase δ in lagging strand synthesis, Rad52/BRCA2 mediator proteins in assisting Rad51 loading, and DNA helicases, e.g., Sgs1/BLM/WRN and Mph1[4,31–33]. Notably, like Dna2, WRN helicase in humans can also be fully activated by the RPA70–NAB polypeptide, which lets us to speculate that a similar mechanism may be employed by human RPA to communicate with enzymes/factors. Hence, our work will shed light on the mechanistic dissection of other functional pairs with RPA.

## Methods

**Yeast strains**. Yeast strains used in this study to study resection and DNA-damage sensitivity are derivatives of JKM139 (ho *MATa hml::ADE1 hmr::ADE1 ade1-100 leu2-3,112 trp1::hisG' lys5 ura3-52 ade3::GAL::HO*). All strains are listed in Supplementary Table 3. The *dna2-AC* mutant was constructed in a *pif1-m2* derivative of JKM139 using in vivo site-directed mutagenesis and confirmed by sequencing[34]. This strain was then back-crossed to JKM179 (MATα derivative of JKM139) to create the single mutant *dna2-AC*.

**Expression and purification of Dna2 proteins**. Yeast cell pellets with Dna2 variants over-expressed, typically 30–40 g, were ground into fine powder in the presence of dry ice and dissolved in 60–80 mL of the resuspension buffer (20 mM Tris-HCl, pH 8.0, 500 mM NaCl, 5% glycerol, 0.01% NP-40, and 1 mM β-mercaptothanol) with 1 mM phenylmethylsulfonyl fluoride (PMSF) and protease inhibitor cocktail (5 μg/ml for each of aprotinin, chymostatin, leupeptin and pepstatin A). The resuspended cell lysate was transferred to the 4 °C cold room for all subsequent purification processes. The lysate is first clarified by centrifugation at 10,000 g for 30 mins. The lysate supernatant was then applied to a self-assembled 5 mL Ni-NTA Fast Flow column (GE Healthcare), washed subsequently with 50 mL of resuspension buffer and 50 mL resuspension buffer with 1 mM ATP and 1 mM MgCl₂. The Dna2 proteins were eluted off the Ni-NTA column by 40 mL resuspension buffer with 250 mM imidazole. The eluent from the Ni-NTA column was incubated with 0.5 mL anti-FLAG-M2 resin (Sigma-Aldrich) overnight. On the following day, the resin was settled in a Bio-Rad Poly-Prep gravity column, washed stepwise with 20 mL resuspension buffer and with 0.1% NP-40, and finally eluted with 10 mL resuspension buffer with 0.25 mg/mL FLAG peptide (30 mins incubation required). The eluent containing purified Dna2 proteins was concentrated in a Millipore Amicon Ultra centricon (5 mL, 30 kDa cutoff) and the FLAG peptide was removed by buffer change with the resuspension buffer. Purified Dna2 proteins were concentrated to 0.5–1 mg/mL and stored in the −80 °C freezer as 10 μL aliquots for future usage. Purified dna2–501 N and dna2–Δ501N–D657A were kept as 20 μL aliquots at 50–100 μg/mL, due to the lower yield.

**Expression and purification of NAB–dna2Δ248 N and NAB–dna2Δ405 N**. The pETduet–NAB–dna2Δ248 N or pETduet–NAB–dna2Δ405 N construct was transformed into *E. coli* Rosetta 2(DE3) pLysS cells to obtain colonies on Luria-Bertani (LB) agar plates with 100 μg/mL ampicillin and then subsequently a single colony was inoculated in 2 L of LB liquid medium with 100 μg/mL ampicillin overnight statically at 37 °C. On the following day, the culture was continued to inoculate at 150 rpm to grow to optical density (OD₆₀₀) of 0.6–0.8, and isopropyl-β-D-10thiogalactopyranoside (IPTG) was added to the final concentration of 0.5 mM to induce protein overexpression at 18 °C for overnight. Cells were harvested by centrifugation at 4300 g for 20 mins and the pellets were stored in the −80 °C freezer.

On the day of protein purification, cell pellets from 2-L culture were thawed and resuspended in 50-mL low-salt buffer (20 mM Tris-HCl, pH 8.0, 100 mM NaCl, 5% glycerol, 0.01% NP-40, and 1 mM β-mercaptothanol) with 1 mM PMSF and protease-inhibitor cocktail (5 μg/ml for each of aprotinin, chymostatin, leupeptin and pepstatin A) and subsequently lysed by sonication on ice for 10 mins. The resuspended cell lysate was transferred to the 4 °C cold room for all subsequent purification processes. The lysate was first clarified by centrifugation at 10,000 g for 30 mins and loaded onto a 2 mL Heparin column (GE Healthcare). The Heparin column was washed with 20 mL low-salt buffer and the bounded proteins were eluted in 20 mL high-salt buffer (20 mM Tris-HCl, pH 8.0, 500 mM NaCl, 5% glycerol, 0.01% NP-40, and 1 mM β-mercaptothanol). The eluate was subsequently loaded onto a 1 mL Ni-NTA column (GE Healthcare), washed with 10 mL high-salt buffer containing 20 mM imidazole, and finally eluted in 5 mL high-salt buffer containing 200 mM imidazole. The eluate from the Ni-NTA column was concentrated to 1 mL with Millipore Amicon Ultra centricon (5 mL, 30 kDa cutoff) and further purified on a Superose 6 10/300 GL gel-filtration column (GE Healthcare). Fractions from the Superose 6 column that contained the NAB–dna2Δ248 N or NAB–dna2Δ405 N target protein were pooled, concentrated to 0.5–1 mg/mL, and stored in the −80 °C freezer as 10 μL aliquots for future usage.

**Construction, expression, and purification of RPA, RPA-ΔN, Rfa1 truncations, and Rfa1–NAB mutants**. The wild-type yeast RPA construct was kindly obtained from Dr. Ilya Finkelstein's lab with *RFA1*, *RFA2*, and *RFA3* genes cloned into the pET11c vector for overexpression in *E. coli*. The RPA-ΔN construct was subcloned subsequently into a pETduet vector with *Nco*I and *Sal*I sites, with the coding sequence for the first 180 amino acids of Rfa1 removed. Both pET11c-RPA and pET11c-RPA-ΔN constructs were transformed into *E. coli* BL21 codon plus cells to obtain colonies on LB agar plates with 100 μg/mL ampicillin, and then subsequently a single colony was inoculated in 2 L LB liquid medium with 100 μg/mL ampicillin overnight statically at 37 °C. On the following day, the culture continued to grow at 150 rpm to an OD₆₀₀ of 0.6–0.8, and IPTG was added to a final concentration of 0.5 mg/mL to induce protein overexpression at 18 °C overnight. Cells were harvested by centrifugation at 4300 g for 20 mins and the pellets were stored in the −80 °C freezer.

Nontagged RPA and RPA-ΔN were purified in a slightly modified protocol following the classic purification scheme for RPA proteins[35]. Specifically, cell pellets from 2 L culture were thawed and resuspended in 50 mL resuspension buffer (20 mM Tris-HCl, pH 8.0, 500 mM NaCl, 5% glycerol, 0.01% NP-40, and 1 mM β-mercaptothanol) with 1 mM PMSF and protease-inhibitor cocktail (5 μg/ml for each of aprotinin, chymostatin, leupeptin and pepstatin A) and subsequently lysed by sonication on ice for 10 mins. The resuspended cell lysate was transferred to the 4 °C cold room for all subsequent purification processes. The lysate was first clarified by centrifugation at 10,000 g for 30 mins and loaded onto a 10 mL self-assembled Affi-Gel Blue Gel gravity column (Bio-Rad). The column was washed sequentially with 50 mL resuspension buffer containing an additional 0.3 M NaCl, 0.5 M NaSCN, and 1.5 M NaSCN. Both RPA and RPA-ΔN eluted in the wash buffer containing 1.5 M NaSCN. The eluent was then applied to a 10 mL self-assembled hydroxyapatite column (Bio-Rad), washed with 50 mL resuspension buffer and eluted in 50 mL phosphate buffer (100 mM KH₂PO₄, pH 8.0, 0.01% NP-40 and 1 mM β-mercaptothanol). At this point, RPA and RPA-ΔN were already 80% pure and thus were concentrated to 1 mL with Millipore Amicon Ultra centricon (5 mL, 30 kDa cutoff) before further purification on a prepacked Superdex 200 10/300 GL gel-filtration column (GE Healthcare). The fractions containing RPA or RPA-ΔN proteins were pooled, diluted to 100 mM NaCl salt strength, and applied to a 1 mL prepacked Mono Q column (GE Healthcare). The mono Q column purification scheme was developed with a salt gradient from

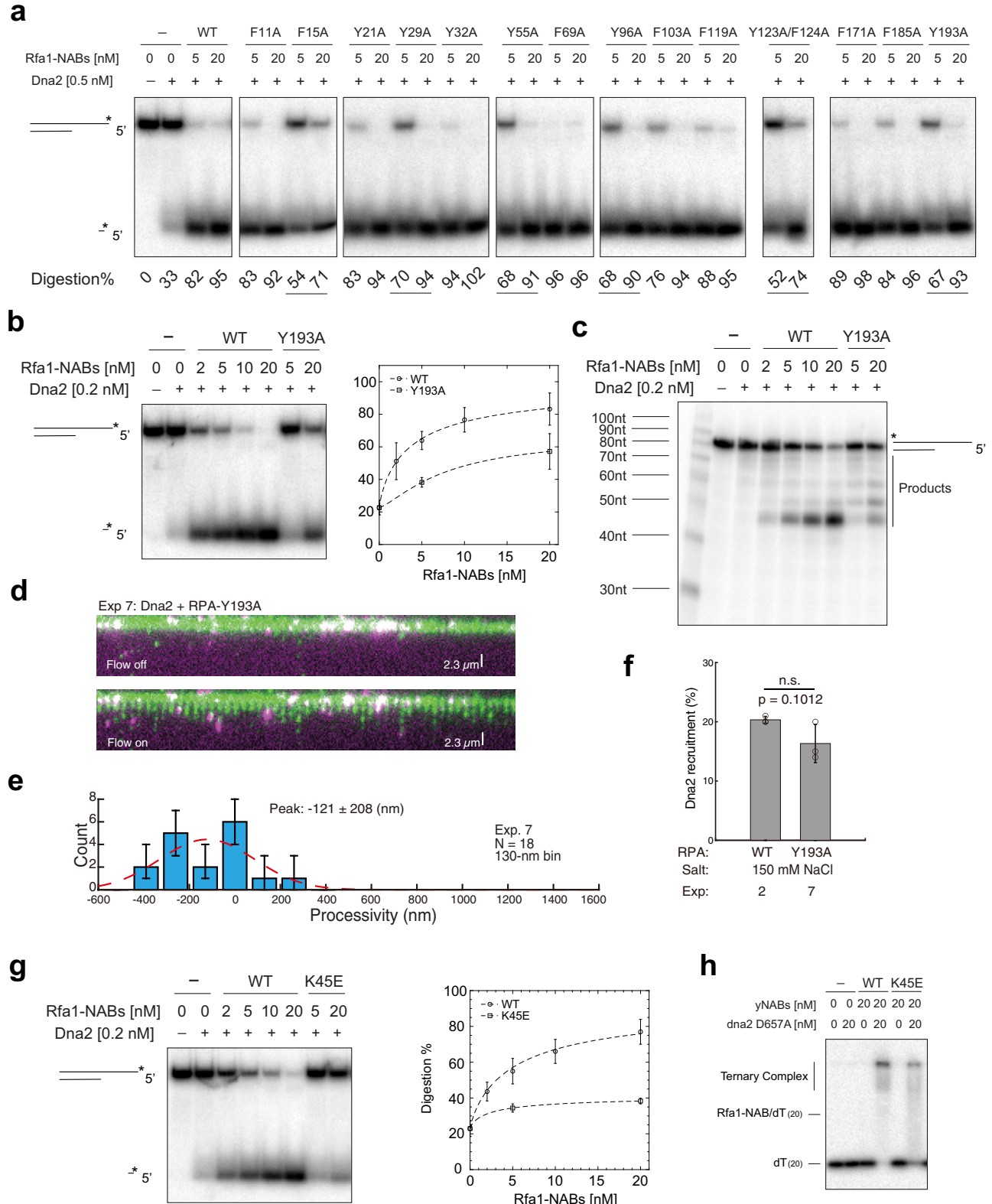

100 mM NaCl to 500 mM NaCl. RPA and RPA-ΔN both eluted between 200 mM and 230 mM NaCl. Purified RPA proteins were slightly concentrated to 2–3 mg/mL and stored in the −80 °C freezer as 10 μL aliquots.

Rfa1-truncation constructs were generated by using the pET11c-RPA construct as the template. The PCR products were cloned into pHis-parallel vector with NcoI and SalI sites to fuse Rfa1 truncations to an N-terminal 6xHis tag, which can be optionally removed by TEV-protease digestion. These constructs generated were pHis–Rfa1 1–435 (NAB), 1–299 (NA), 1–180 (N), and 181–435 (AB). The ssDNA-binding mutants of Rfa1–NAB were generated by site-directed mutagenesis using

pHis–Rfa1 1–435 as the template and these constructs generated were pHis-Rfa1 1-435 F238A/F269A (NA⁻B), W360A/F385A (NAB⁻), and F238A/F269A/W360A/F385A (NA⁻B⁻). The mutants from Rfa1-N were also generated by site-directed mutagenesis using pHis–Rfa1 1–435 as the template and these constructs generated were pHis-Rfa1 1-435 F11A, F15A, Y21A, Y29A, Y32A, Y55A, F69A, Y96A, F103A, F119A, Y123A/F124A, F171A, F185A, Y193A, and K45E.

All the Rfa1-truncation proteins were purified using Ni-NTA chromatography, followed by gel-filtration chromatography. Specifically, the constructs were transformed into *E. coli* BL21 codon plus cells, and cell pellets containing

**Fig. 5 Tyrosine 193 in Rfa1 is critical for processive ssDNA digestion by the Dna2–RPA ensemble. a** Digestion by Dna2 on 5′-labeled 40-nt 5′-overhanging ssDNA (5 nM) with Rfa1–NAB and Rfa1–NAB aromatic mutants. Percentage of digestion from one-time screening experiments was listed with partially defective mutants underlined. **b** Digestion by Dna2 on 5′-labeled 40-nt 5′-overhanging ssDNA (5 nM) with Rfa1–NAB and rfa1–NAB–Y193A mutant. For quantification, mean values ± s.d. from three independent experiments were plotted. **c** Digestion by Dna2 on 3′-labeled 40-nt 5′-overhanging ssDNA (5 nM) with Rfa1–NAB and rfa1–NAB-Y193A mutant. The experiments were repeated three times. **d** Wide-field image of a Dna2 (magenta punctum)–RPA-Y193A-GFP complexes digesting the ssDNA–RPA–Y193A-GFP complexes (green) (Exp. 7 in Supplementary Table 1). **e** Processivity distribution of single Dna2 digestion with RPA-Y193A in solution (Exp. 7 in Supplementary Table 1): 130-nm bin. Digestion events $N = 18$. Each experimental condition was examined over more than three DNA Curtains experiments ($n \geq 3$). Error bars were obtained through the bootstrap analysis. For any normally distributed dataset, 68.27% of the values lie within one standard deviation of the mean, therefore, our choice of 70% confidence intervals for the bootstrapped data provides a close approximation to expectations for one standard deviation from the mean. The data were fitted with Gaussian functions (red dash line). The errors represented 95% confidence intervals obtained through Gaussian function fitting. **f** Dna2 recruitment (%) with RPA-Y193A in solution (Exp. 7 in Supplementary Table 1). Independent DNA Curtains experiments were repeated: $n = 3$ for Exp. 2; $n = 3$ for Exp. 7; Error bars, mean ± s.d. Statistical significance was analyzed using the unpaired t-test for two groups. p-value: two-tailed; p-value style: GP: 0.1234 (ns), 0.0332 (*), 0.0021 (**), 0.0002 (***), <0.0001 (****). Confidence level: 95%. **g** Digestion by Dna2 on 5′-labeled 40-nt 5′-overhanging ssDNA (5 nM) with Rfa1–NAB and rfa1–NAB-K45E mutant. For quantification, mean values ± s.d. from three independent experiments were plotted. **h** Formation of ternary complex from dna2–D657A, Rfa1–NAB variants, and $dT_{(20)}$ ssDNA (5 nM). The experiments were repeated three times. Source data are provided as a Source Data file.

overexpressed proteins were obtained similarly as the wild-type RPA. After cell lysis by sonication, the clarified cell lysate from a 2 L culture was loaded onto a 5 mL Ni-NTA gravity column (GE Healthcare) and then washed sequentially with 50 mL resuspension buffer containing 20 mM imidazole and eluted with the same buffer also containing 250 mM imidazole. The eluent was concentrated to 5 mL using a Millipore Amicon Ultra centricon (5 mL, 10 kDa cutoff) before further purification with a 120 mL self-packed Superdex 200 gel-filtration column (GE Healthcare). Peak fractions were combined and stored in the −80 °C freezer as 1 mL aliquots at 1–5 mg/mL.

**Construction, expression, and purification of hRPA, hRPA70 truncations, and yN-hAB hybrid.** The wild-type human RPA construct was also kindly obtained from Dr. Ilya Finkelstein's lab which contains *RPA70*, *RPA32*, and *RPA14* genes cloned into the pET11c vector for over-expression in *E. coli*. Nontagged hRPA was over-expressed and purified with the same scheme as yeast RPA and saved at 5 mg/mL in the −80 °C freezer as 10 µL aliquots.

hRPA70-truncation constructs were generated with the same subcloning scheme as Rfa1-truncation constructs. They were pHis–hRPA70 1–435 (hNAB), 181–435 (hAB), and 1–180 (hN). The yN–hAB construct was generated by fusing the yN domain (Rfa1 1–180) to the hAB domain (hRPA 181–435) by a two-step overlapping PCR scheme and the pHis–yN–hAB construct was verified by DNA sequencing. These hRPA70 truncation proteins and the yN–hAB protein were subsequently purified in the same scheme as yRfa1 truncation proteins and saved as 1 mL aliquots at 1–5 mg/mL in the −80 °C freezer.

**Preparation of DNA substrates.** All oligonucleotides were purchased from IDT. The sequence of oligonucleotides (S1–S14) other than poly-dT is listed in Supplementary Table 4. For the 5′-end labeling, the oligonucleotide was radiolabeled with [γ-32P] ATP using T4 Polynucleotide Kinase (New England Biolabs). For the 3′-end labeling, the oligonucleotide was radiolabeled with [α-32P] dATP using terminal transferase (New England Biolabs). After the labeling reactions, excess ATP/dATP was removed with Bio-Spin 6 columns (Bio-Rad) before annealing to the corresponding oligonucleotides in New England Biolabs Buffer 3.1 (50 mM Tris-HCl, pH 7.9, 100 mM NaCl, 10 mM MgCl₂, and 100 µg/mL BSA) using a standard protocol performed in a thermal cycler. To label long ssDNA/dsDNA substrates internally, the [α-32P] dATP was directly added to the PCR reaction. The PCR product was agarose-gel-purified and kept at −20 °C in the form of dsDNA. On the day of the assay, the ssDNA substrate was made by diluting the dsDNA to the desired concentration, heating at 95 °C for 10 min, and immediately cooling on ice for 10 min, right before the reactions.

For Dna2 nuclease assays, S1 was annealed with S2 and S3 individually to create 40-nt and 60-nt 5′-overhang substrates respectively. S4 was annealed with S5, S6, S7, and S8 individually to generate 80-nt, 100-nt, 120-nt, and 140-nt 5′-overhanging ssDNA, respectively. S9 was annealed with S8 to create the 140-nt 5′-overhanging ssDNA substrate with a hairpin in the ssDNA region. To generate long ssDNA substrate for Dna2 nuclease assays, a part of the Rfa1 coding sequence (562 bp) was PCR-amplified and radiolabeled either internally or at its 5′-end. ssDNA was prepared as described above immediately before assays. The 600-bp dsDNA substrate used in resection assays was PCR amplified using the yeast Ku80 open-reading frame (ORF) as the template. For electrophoretic mobility-shift assays (EMSA), S10 was annealed with S11 to generate the 5′-overhanging $dT_{(30)}$ ssDNA substrate. S12 was annealed with S13 to generate the 3′-overhanging $dT_{(30)}$ ssDNA substrate. Finally, S14 was annealed with S11 and S13 to generate the $dT_{(30)}$-gap ssDNA substrate.

**Dna2-nuclease assays.** Unless specified, Dna2-nuclease assays were conducted with 150 mM KCl. We note that although we are trying to match the physiological salt concentration, neither the KCl used in our nuclease assays nor the NaCl used in the DNA Curtains analysis truly represents the physiological salt in cells, as the concentration of Cl ion in yeast is generally maintained at less than 1 mM, regardless of the NaCl or KCl concentration outside the cells. For short substrates, Dna2 truncations and Dna2 mutants, 0.2 nM or 0.5 nM Dna2 and its variants were incubated without or with RPA variants and 5 nM 5′-overhang substrates radiolabeled at either the 5′-end or the 3′-end, in 10 µL reaction buffer (25 mM Tris-HCl, pH 7.5, 150 mM KCl, 2 mM MgCl₂, 1 mM DTT, and 100 µg/mL BSA). The reactions were initiated by the addition of substrate DNA, incubated at 30 °C for 10 min, and terminated by moving the reactions on ice, followed by deproteinization with SDS-PK (1.2% SDS and 3 mg/mL proteinase K) treatment at 37 °C for 10 min. The samples were resolved either on a native 10% polyacrylamide gel in 0.5x TBE buffer (Tris-Boric Acid EDTA) or on a 12% denaturing polyacrylamide gel in 0.5x TBE buffer after heat-denaturing at 95 °C for 10 min. When Dna2 variants were examined for the digestion of long ssDNA substrate, reactions were set up similarly to the short substrate, except that 1 nM Dna2 variants were typically used. To assess the salt impact on Dna2 nuclease activity, the salt concentration in the reaction buffer was adjusted to either 50 mM KCl or 250 mM KCl as indicated.

**Electrophoretic mobility-shift assays (EMSA).** In EMSA, dna2–D657A was incubated with 5 nM 5′-$dT_{(30)}$-overhanging ssDNA, 3′-$dT_{(30)}$-overhanging ssDNA and $dT_{(30)}$-gap ssDNA substrates in 10 µL reaction buffer (25 mM Tris-HCl, pH 7.5, 150 mM KCl, 1 mM DTT, and 100 µg/mL BSA) with the indicated amount of RPA. The reaction was initiated by the addition of ssDNA substrates, kept at 30 °C for 10 min, and terminated by moving to room temperature. Samples were immediately resolved on a 4% native polyacrylamide gel in 0.2x TBE buffer cooled on ice. When the ternary-complex formation between Dna2 and RPA was analyzed by the EMSA assays, dna2–Δ405N–D657A and dna2–Δ501N–D657A were also included along with dna2–D657A in order to incubate with various forms of Rfa1 truncations and mutants. In all, 5′-end radiolabeled poly-dT oligos of various lengths were tested for the ternary-complex formation with Dna2 and RPA variants.

**Affinity pull-down assays.** To explore the interaction between Dna2 and RPA, 2 µg of purified FLAG–His₆-tagged Dna2 or dna2-AC was incubated with 2 µg of RPA in 30 µL interaction buffer (20 mM KH2PO4, pH 7.4, 10% glycerol, 0.5 mM EDTA, 0.01% NP-40, 1 mM DTT, 15 mM imidazole, and 50 mM KCl) at 4 °C for 30 min. A reaction with only 2 µg of RPA present was also set up as a control. In all, 10 µL of Ni-NTA resin (Qiagen) was applied to each reaction mix and the mixtures were further incubated at 4 °C for 30 min. After the incubation, the supernatant was removed and the resin was washed three times with 200 µL reaction buffer containing 0.1% NP-40. Bound proteins on the resin were eluted with 15 µL of 2% SDS. The supernatant (S), wash (W), and SDS eluate (E), 10 µL each, were analyzed by 10% SDS-PAGE.

**Microscale-thermophoresis (MST) assays.** To determine the disassociation constants between Dna2 and RPA, either RPA or His–FLAG-tagged Dna2 and dna2-AC, was buffer-exchanged into MST-reaction buffer (20 mM Tris-HCl, pH 8.0, 150 mM NaCl, 5% glycerol, 0.01% NP-40, and 1 mM β-mercaptothanol) before analysis on a Microscale Thermophoresis Nanotemper Monolith NT.115 Blue/Red, which was controlled by NT Control software from Nano Temper Technologies. Dna2 and dna2-AC proteins were fluorescently labeled using the His-tag labeling kit

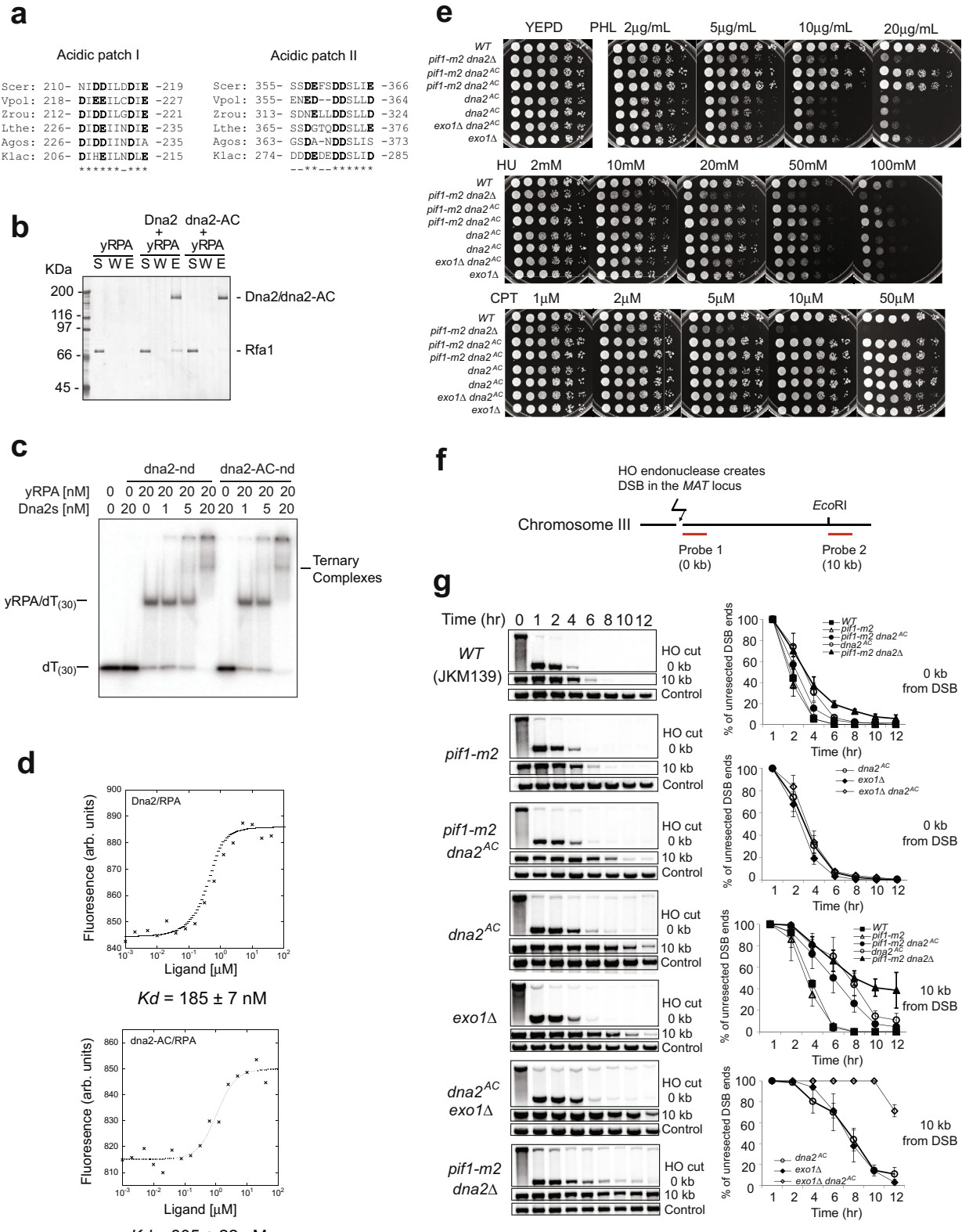

RED-tris-NTA, and RPA was titrated as ligand to determine the Kd using hydrophilic capillaries at 30 °C. The Kd values for MST assays were determined in Kd Fit mode in NT Analysis software also from Nano Temper Technologies.

**Cross-linking assays**. For cross-linking experiments, Dna2 and RPA proteins were purified using a phosphate-based buffer system (pH 7.4), instead of the Tris-HCl buffer system, to avoid non-protein sourced primary amine groups in the

cross-linking mixture, which can react with the cross-linking reagent, BS[3] (bis[-sulfosuccinimidyl] suberate, Thermo Fisher). To assess the interface between Dna2 and RPA in the presence of ssDNA substrate, Dna2 was incubated with RPA, with both present in solution at 1 µM, at room temperature for 30 min in the absence or the presence of $dT_{(30)}$ ssDNA (0.5–5 µM), and in 25 µL crosslinking buffer (20 mM HEPES, pH 7.5, 150 mM NaCl, and 400 µM BS[3]). A separate group with 30 µM dT replacing the $dT_{(30)}$ ssDNA was also set up. After incubation, the cross-linking experiments were quenched by the addition of 2 µL 1 M Tris-HCl, pH 8.0, and the

**Fig. 6 dna2-AC, an RPA interaction-defective mutant of Dna2, supports cell viability, but sensitizes cells to phleomycin. a** Protein-sequence alignment for the acidic patch-I and acidic patch-II regions from Dna2 proteins across multiple fungal species. (Scer: *Saccharomyces cerevisiae*; Vpol: *Vanderwaltozyma polyspora*; Zrou: *Zygosaccharomyces rouxii*; Lthe: *Lachancea thermotolerans*; Agos: *Ashbya gossypii*; Klac: *Kluyveromyces lactis*). **b** Affinity pulldown utilizing nickel-NTA resin to examine the interaction of (His)$_6$-tagged Dna2 and dna2-AC with RPA. The supernatant (S), wash (W), and eluate (E) fractions were analyzed by SDS-PAGE. The experiments were carried out once. **c** Formation of ternary complex from titrated Dna2 variants (0–20 nM), RPA (20 nM), and dT$_{(30)}$ ssDNA (5 nM). The experiments were repeated three times. **d** Equilibrium-dissociation constants (*Kd*) between RPA and Dna2 or dna2-AC were measured by microscale-thermophoresis assays to be 185 ± 7 nM and 805 ± 22 nM respectively. The experiments were repeated three times. **e** Analysis of sensitivity to phleomycin, hydroxyurea, and camptothecin in the indicated mutants. Exponentially growing cells were spotted at 5-fold serial dilutions on plates supplemented with DNA-damage reagents. The plates were scanned after 3~4 days of incubation at 30 °C. **f** Simplified scheme for HO endonuclease assay for evaluation of cellular resections from *Saccharomyces cerevisiae*. **g** Southern blot analysis of resection kinetics at 0 and 10 kb from the DSB in indicated mutants. For quantification, mean values ± s.d. from three independent experiments were plotted. Source data are provided as a Source Data file.

samples were subsequently analyzed with 4–15% gradient SDS-PAGE. In a similar strategy, Dna2 and RPA were incubated in the presence of a series of poly-dT ssDNA (dT$_{(5)}$, dT$_{(10)}$, dT$_{(15)}$, dT$_{(20)}$, dT$_{(30)}$, and dT$_{(60)}$) at 1 μM or 10 μM in cross-linking buffer, and the reactions were quenched and analyzed with 4–15% gradient SDS-PAGE.

**Mass spectrometry**. Gel bands corresponding to the cross-linked proteins were diced into 1 mm cubes. Proteins were reduced with 10 mM TCEP at 56 °C for 45 min, and then alkylated with 20 mM iodoacetamide at 21 °C for 30 min in the dark. Proteins were digested with trypsin at 12.5 ng/μL for 16 hr. The resulting peptides were desalted using a C18 ZipTip (Millipore). Tryptic peptides were injected into an Easy-nLC 1000 HPLC system coupled to an Orbitrap Fusion Lumos mass spectrometer (Thermo Scientific), which was controlled by Thermo Scientific Xcalibur software version 4.1. Peptide samples were loaded onto an Acclaim PepMap$^{TM}$ 100 C18 trap column (75 μm × 20 mm, 3 μm, 100 Å) in 0.1% formic acid. The peptides were separated using an Acclaim PepMap$^{TM}$ RSLC C18 analytical column (75 μm × 150 mm, 2 μm, 100 Å) using an acetonitrile-based gradient (Solvent A: 0% acetonitrile, 0.1% formic acid; Solvent B: 80% acetonitrile, 0.1% formic acid) at a flow rate of 300 nL/min. A 30 min gradient was as follows: 0–1.5 min, 2–10% B; 1.5–31.5 min, 10–45% B; 31.5–35 min, 45–100% B; 35–40 min, 100% B, followed by re-equilibration to 2% B. The electrospray ionization was carried out with a nanoESI source at a 260 °C capillary temperature and 1.9 kV spray voltage. The mass spectrometer was operated in a data-dependent acquisition mode with a scan range of 400–2000 m/z at the resolution of 120,000. The auto gain-control target was set at $4 \times 10^5$. The precursor ions with charge states from 3 to 6 were selected for tandem mass (MS/MS) analysis in Orbitrap using HCD and CID at 35% collision energy. The intensity threshold for MS2 was set at $5 \times 10^4$ and the Orbitrap resolution was set to 30,000. The dynamic exclusion was set with a repeat count of 1 and an exclusion duration of 10 seconds.

**Mass-spectrometry data analysis**. The resulting mass spectrometry data were utilized to perform database search in Protein Prospector (http://prospector.ucsf.edu/prospector/mshome.htm) against Dna2 and RPA sequences. Carbamidomethylation of cysteine residues was set as a fixed modification. Protein N-terminal acetylation, oxidation of methionine, protein N-terminal methionine loss, and pyroglutamine formation were set as variable modifications. Cross-linking search type was set as DSS (water-insoluble analog of BS$^3$, which has the same mass addition). A total of three variable modifications were allowed. Trypsin-digestion specificity with two missed cleavages was allowed. The mass tolerance for precursor and fragment ions was set to 5 ppm.

As identifications of cross-linked peptides are only considered correct if both peptide components are identified, crosslink identification confidence is largely a function of the less-confident peptide. The resulting cross-linked peptides were manually validated and categorized into three groups as listed in Supplementary Table 2: high confidence (designated as Rank 1), low confidence (designated as Rank 2), and mass-alone (designated as Rank 4). The high-confidence group included at least three unique y- or b-type fragment ions derived from the lower-scoring peptide of the pair. The low-confidence group included cross-linked peptides whose lower-scoring peptide had only one or two unique fragment ions. The mass-alone group meant the lowest-scoring cross-linked peptide had fewer than six amino acid residues, such that there is a much-reduced chance of observing three unique fragment ions. The CID results, which were not illustrated in the cross-linking maps, were used for further validation of HCD results if the same cross-linked peptide pairs were identified in both fragmentation methods. The cross-linking maps were generated with XiNET (http://crosslinkviewer.org/index.php)[36].

**Bootstrap analysis**. Matlab function 'bootci' (bootstrap confidence interval, R2015a; MathWorks, Inc., Natick, MA) was used to calculate the error bars for position distributions in Figs. 1e–i, 2a, and d, 5d–f, and 7c–h, and Supplementary Fig. 2g–j representing 70% confidence intervals. Another parameter, repeat =1000, which is the number of bootstrap samples used in the computation (nboot).

**Unpaired *t*-test**. Statistical significance was evaluated based on Student's *t*-tests (Prism 9 for macOS, Version 9.1.0 (216), March 15, 2021, GraphPad Software, Inc.). The test was chosen as unpaired *t*-test. *P*-value style: GP: 0.1234 (ns), 0.0332 (*), 0.0021 (**), 0.0002 (***), < 0.0001 (****).

**Preparation of long dsDNA substrates for DNA Curtains**. We conducted PCR to prepare the human beta-globin dsDNA substrates (29,951 bp). The PCR kit was Promega GoTaq Long PCR GoTaq Master mix (Cat. # M4021L). The template was the human genome (Hela cell). Both designed primers were (1) 5′-[Biotin]-TGCTGCTCTGTGCATCCGAGTG-3′, (2) 5′-AGCTTCCCAACGTG ATCGCCTTTC TC-3′. The final PCR product of DNA substrates contained a 5′-Biotin, which was used to tether the substrate to the flowcell surface of DNA Curtains[17,18].

**Total internal-reflection fluorescence microscope (TIRFM)**. All experimental data of DNA Curtains were acquired with a custom-built prism-type two-color TIRFM (Nikon, Inverted Microscope Eclipse Ti-E), and the exposure time was 100-ms. The microscope was mounted with both 488-nm and 561-nm OBIS LS 100-mW lasers. The real laser powers before a prism were measured: (i) 488 nm, 7.0 mW (10%); and (ii) 561 nm, 26.3 mW (40%).

**Anti-FLAG quantum dot (QD)**. Anti-FLAG antibody (Sigma, #F1804-1MG) was labeled with QD$^®$705 by a commercial kit (Invitrogen, SiteClick$^{TM}$Qdot$^{TM}$705 Antibody Labeling Kit, #S10454).

**The experimental protocol of DNA Curtains**. We first flashed a BSA buffer containing the biotin-labeled DNA substrates (29,951 bp, Supplementary Fig. 2a) above into a flowcell of DNA Curtains for 10 min with a flow rate of 0.2 ml/min. The BSA buffer was 20 mM Tris-HCl (pH 7.5), 2 mM MgCl$_2$, 1 mM DTT, 0.2 mg/ml BSA, and 1 nM YOYO-1. YOYO-1 was used to stain DNA. Afterward, DNA Curtains were visualized by changing the flow rate to 0.4 ml/min (Supplementary Fig. 2b–c). The length of DNA substrates was measured as 7879 ± 453 nm (Supplementary Fig. 2d). However, the length of B-form dsDNA containing 29,951 bp could be calculated as 29,951 bp × 0.34 nm/bp = 10,183 nm. This result confirmed that the 0.4 mL/min flow rate can extend the substrate to 80% of the total length.

Second, we injected a 50 μL Exonuclease III sample (New England Biolab #M0206L, 0.5 U) into the flowcell for 5-min incubation at 37 °C. The working buffer was changed to New England Biolab Buffer 1: 10 mM Tris-HCl (pH 7.0), 10 mM MgCl$_2$, 1 mM DTT, and 0.2 mg/ml BSA. During the incubation, Exo III can start to digest the dsDNA from one 3′-end to generate a ssDNA substrate with a free 5′-end (Supplementary Fig. 2e). After the incubation, we flashed the BSA buffer containing 0.1 nM RPA–GFP or RPA–ΔN–GFP into the flowcell with a flow rate of 1 ml/min (Supplementary Fig. 2f). During this time, we changed the temperature from 37 °C to 30 °C. The length of these new constructs was measured as 30 ± 1 pixels (Supplementary Fig. 2g), which was 8010 ± 267 nm. Here 1 pixel = 267 nm for the DNA Curtains setup.

During the second step, we also started to prepare a 100 μl Dna2 sample for the third step below. In all, 1.0 pmole anti-FLAG QDs were pre-incubated with 0.2 pmole Dna2 containing a FLAG tag (wild type, dna2-AC, or dna2-D657A) on ice for 10 min. The total reaction volume was 2–3 μl. After the incubation, the sample mixture was diluted with a Dna2 working buffer to 100 μl. The final concentrations of Dna2, RPA, and QD were 2 nM, 2 nM, and 10 nM, respectively. The Dna2 working buffer containing 0.1 nM RPA–GFP or RPA–ΔN–GFP was 20 mM Tris-HCl (pH 7.5), 2 mM MgCl$_2$, 1 mM DTT, 50 mM or 150 mM NaCl, and 0.2 mg/ml BSA.

For the third step, we flashed the 50 μl Dna2 sample into the flowcell with the Dna2 working buffer for 15-min (Fig. 1c). The 488-nm and 561-nm lasers were turned on to start the data acquisition with a 100-ms exposure time and 5-s intervals. The flow rate was 1.0 mL/min. The starting time point was defined as the time when Dna2 was observed binding to the ssDNA–RPA complex.

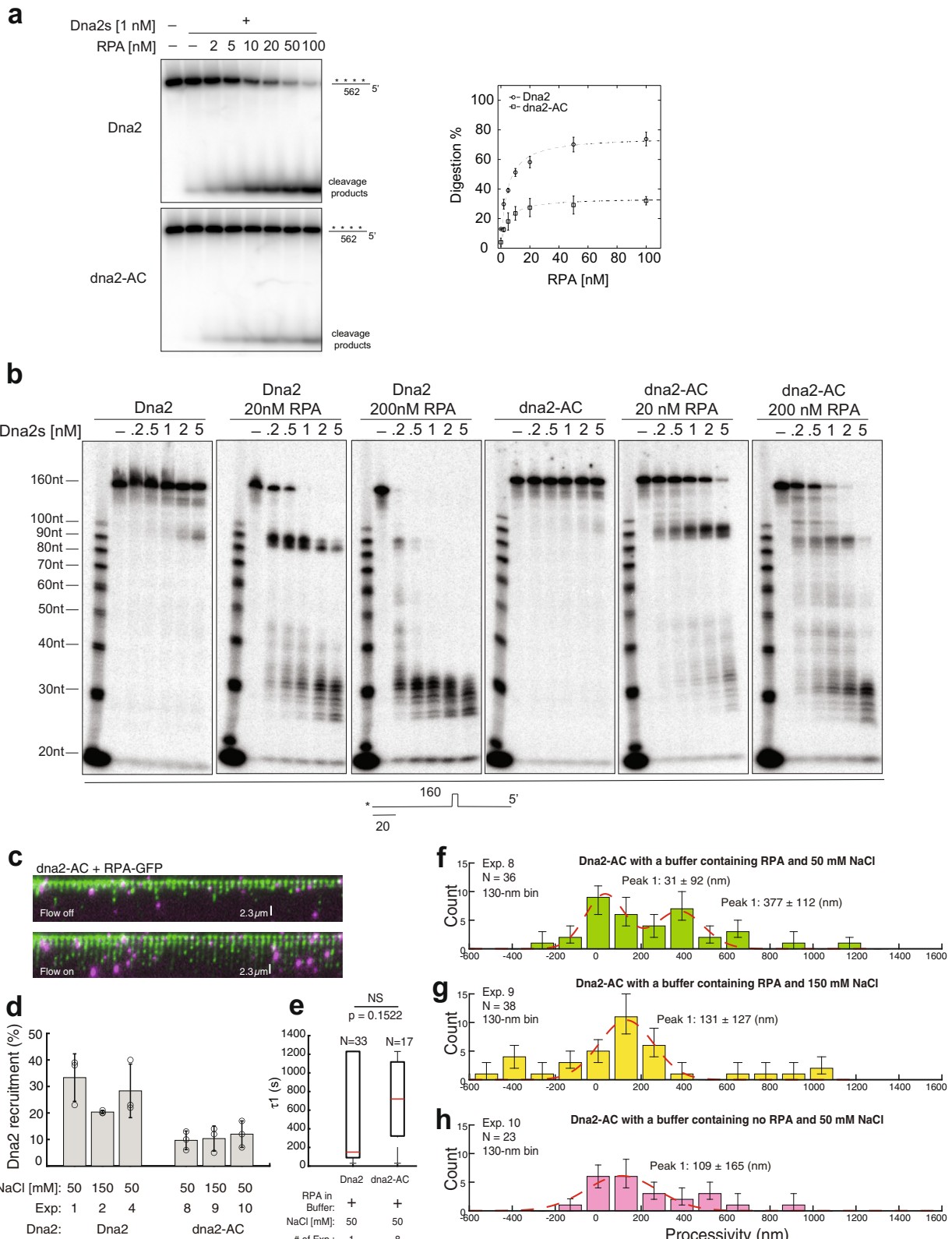

**Image tracking for DNA Curtains experiments**. QD-labeled Dna2 (magenta puncta) molecules in DNA Curtains experiments digested the 5′-free ssDNA–RPA complexes. Image analysis was performed using open-source image-processing software ImageJ (Version: 2.0.0-rc-59/1.51k, http://imagej.net/Contributors). A particle-tracking script: "Speckle TrackerJ" (version: 0.87 built:10/11/2011)" (http://athena.physics.lehigh.edu/speckletrackerj) was used to track the magenta puncta image of Dna2. "Diffusing_ spots" model was chosen for tracking, and "Gaussian_fit" was chosen for 2D-Gaussian fitting.

**One nucleotide (nt) ~0.27 nm**. Accurate conversion between the length unit (nm) and the number of nucleotides (nt) is difficult, and here we provide a simple estimation. The average length of DNA substrates after Exo-III digestion was $30 \pm 1$ pixels (Supplementary Fig. 2g), which was $8010 \pm 267$ nm. Here 1 pixel = 267 nm for DNA Curtains setup. The maximum number of the final ssDNA substrate after Exo-III digestion was 29,951-nt, and thus 29,951-nt $\leq 8010 \pm 267$ nm. From this relationship, we get $1$ nt $\leq 0.27 \pm 0.01$ nm.

**Fig. 7 ssDNA digestion by dna2-AC, is less processive and sensitive to DNA structures. a** Impact of titrated RPA on the digestion of internally labeled 562-nt ssDNA (5 nM) by Dna2 or dna2-AC respectively. For quantification, mean values ± s.d. from three independent experiments were plotted. **b** Digestion by titrated Dna2 and dna2-AC, on 3'-labeled 140-nt 5'-overhanging ssDNA (5 nM) (hairpin-containing) without or with RPA. The experiments were repeated three times. **c** Wide-field image in DNA Curtains experiments of dna2-AC (magenta puncta) digesting the RPA–ssDNA complexes (green). **d** Dna2 recruitment (%) for different experimental conditions (Supplementary Table 1). Independent DNA Curtains experiments were repeated: $n = 3$ for Exp. 1, 2, 4, 8, 9, 10. Error bars, mean ± s.d. **e** Boxplot of $\tau1$ for different experimental conditions (Supplementary Table 1). The total number of Dna2 digestion events in different experimental conditions examined over more than three DNA Curtains experiments ($n \geq 3$) was indicated in all boxplots. For the boxplot, the red bar represents the median. The minima of the box represents 25th percentiles, and the maxima is 75th percentiles. Most extreme data points are covered by the whiskers except outliers. The '+' symbol is used to represent the outliers. Statistical significance was analyzed using the unpaired $t$-test for two groups. $p$-value: two-tailed; $p$-value style: GP: 0.1234 (ns), 0.0332 (*), 0.0021 (**), 0.0002 (***), <0.0001 (****). Confidence level: 95%. **f–h** Processivity distribution of single DNA2-AC digestion for different experimental conditions (Supplementary Table 1): 130-nm bin. The total number of dna2-AC digestion events (N) was as labeled. Each experimental condition was examined over more than three DNA Curtains experiments ($n \geq 3$). Error bars were obtained through the bootstrap analysis. For any normally distributed dataset, 68.27% of the values lie within one standard deviation of the mean, therefore, our choice of 70% confidence intervals for the bootstrapped data provides a close approximation to expectations for one standard deviation from the mean. The data were fitted with Gaussian functions (red dash line). The errors represented 95% confidence intervals obtained through Gaussian function fitting. Source data are provided as a Source Data file.

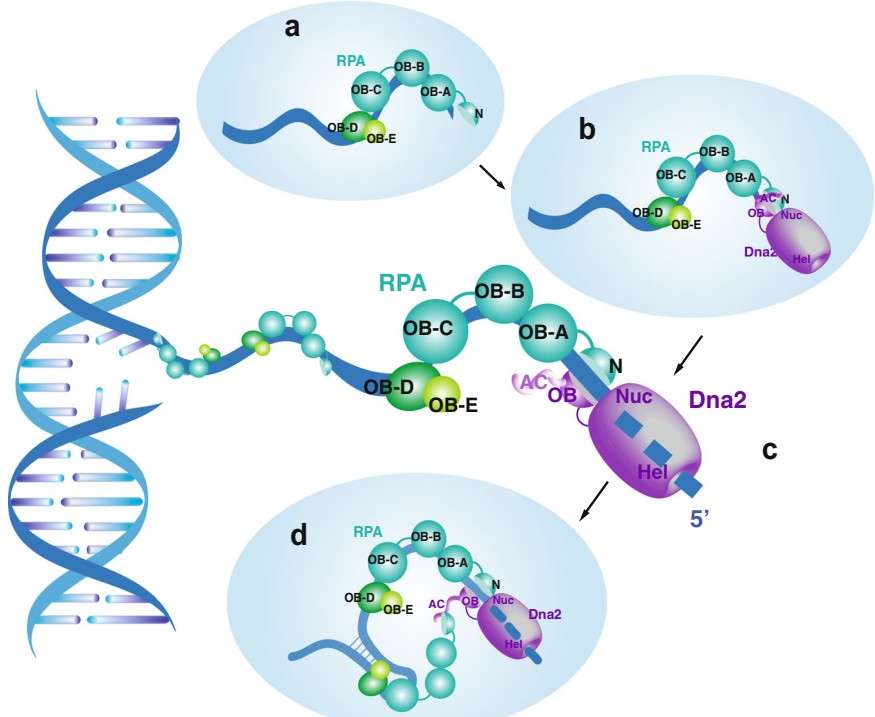

**Fig. 8 A model of ssDNA digestion by the Dna2–RPA ensemble. a** RPA coats ssDNA. **b** RPA recruits Dna2 to the open 5'-end of ssDNA. **c** RPA acts as a processive unit for Dna2 and delivers ssDNA for its digestion. **d** Dna2 trans-acts with RPA via its AC motif to overcome DNA structures.

**Analysis of resection at DSB ends**. Resection was analyzed at an HO endonuclease-induced nonrepairable DSB at the *MAT*a locus on chromosome III using Southern blots as described[10]. Briefly, genomic DNA was isolated by glass-bead disruption using a standard phenol-extraction method. Purified DNA was digested with *Eco*RI and separated on a 0.8% agarose gel. Southern blotting and hybridization with radiolabeled DNA probes was performed as described previously[37]. Intensities of target bands on Southern blots were analyzed with ImageQuant TL (Amersham Biosciences) and normalized to the *TRA1* probe. Resection efficiency at each time point was calculated from normalized intensities of the bands corresponding to EcoRI restriction fragments detected with probes located either next to the DSB (*MAT*) or 10 kb upstream of the DSB (*SNT1*). Analysis was performed three times in each mutant strain.

**Reporting summary**. Further information on research design is available in the Nature Research Reporting Summary linked to this article.

## Data availability
The data that support this study are available from the corresponding authors upon reasonable request. The mass-spectrometry proteomics data have been deposited to the ProteomeXchange Consortium via the PRIDE[38] partner repository with the dataset identifier PXD028637. Source data are provided with this paper.

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

## Acknowledgements

We are grateful for Drs. Steven Brill, Ilya Finkelstein, and Patrick Sung for providing plasmids, Hayley Bedwell and Alex Chern for technical support, Dr. Marc Wold for the discussion during the development of this project, and physical biochemistry instrumentation facility (PBIF) at Indiana University for equipment support. This work was supported by NIH research grant GM124765 and American Cancer Society Research Scholar award RSG-21-013-01-DMC to H.N., NIH research grant GM080600 and GM125650 to G.I. This work was also supported by NSFC Grant No. 31670762 to Z.Q.

## Author contributions

Z.Q. and H.N. conceived and designed the research. J.S. performed biochemical experiments with Y.L. assistance on the crosslink experiments. Y.Zhao. performed single-molecule DNA Curtains experiments and N.P. performed genetic analysis. Y.Zhang. and J.T. performed mass-spectrometry analyses of the cross-linked samples. J.S., Y.Zhao., N.P., G.I., Z.Q. and H.N. analyzed the data and wrote the paper. All authors commented on the paper.

## Competing interests

The authors declare no competing interests.
