## [Peer Review File · Nature Communications]

Deciphering the mechanism of processive ssDNA digestion by the Dna2-RPA ensembleReviewers' Comments:

Reviewer #1:

Remarks to the Author:

The conserved DNA Flap endonuclease Dna2 fulfils a multi-faceted role in DNA replication and repair. Dna2 degrades both 3' and 5' DNA Flaps, but the single-strand DNA binding protein RPA attenuates 3' Flap cutting while enhancing 5' Flap incision. In DNA replication, Dna2 digests long DNA Flaps generated by the Pif1 helicase and DNA polymerase delta. Dna2 deficiency therefore leads to prolonged DNA damage checkpoint activation and eventual cell death. In DNA double-strand break (DSB) repair by homologous recombination (HR), Dna2 digests 5' Flaps generated by the Sgs1 helicase in conjunction with RPA. This "DNA end resection" reaction produces a 3'-tailed DNA template for the recruitment of Rad51, the general recombinase capable of homologous DNA pairing and strand exchange. Even though we have known for quite some time (almost two decades) that Dna2 functionally co-operates with RPA, how RPA stimulates 5' Flap incision but protects the 3' Flap from being digested remains to be elucidated.

This study employs a variety of biochemical, genetic, and single-molecule biophysical methods to shed light on Dna2-RPA synergy. Evidence is furnished that RPA regulates Dna2 processivity. Importantly, the authors document "trans" synergy between Dna2 and RPA that is dependent on a novel "AC motif" in Dna2. Interestingly, a mutation in the AC motif creates a separation-of-function dna2 mutant that is viable but defective in HR intermediate processing. Overall, this study furnishes insights into a hitherto obscure mechanism germane for understanding the processing of HR intermediates and mechanisms of genome maintenance. The analyses have been conducted exceptionally well and the conclusions are supported by the results.

I have just a few suggestions for the authors to consider, as follows.

1. The Introduction can be expanded to better discuss the mechanisms of Okazaki fragment maturation and DNA end resection.
2. In Figure 1, the processivity of the RPA-Dna2-ssDNA ensemble is likely independent of the helicase activity of Dna2 as no ATP was present. While the motor activity of Dna2 has been well-characterized in bulk biochemical assays, it would be interesting to see whether the addition of ATP has any impact on Dna2 processivity in the DNA curtain analysis.
3. In Figure 3A, the mapped interface between Dna2 and RPA largely resides at the N-terminus and the OB-fold region of Dna2. There is a well-known dna2-2 allele with the P504S mutation. The P504 residue is located right at the hinge area connecting the OB and the nuclease domains of Dna2. It would be interesting to ask whether the P504S mutation affects Dna2-RPA-ssDNA ternary complex formation.
4. The N-terminus of RPA is suggested by the authors to be crucial for complex formation. Besides dna2 mutants, it would be good to also test a Rfa1-N mutant, for instance, the rfa1-t11 mutant.
5. In Figure 3D, the hybrid NAB-Dna2 Δ 248N protein provides a tool to secure cis while voiding the trans action. The authors could also test NAB-Dna2 Δ 248N-AC or NAB-Dna2 Δ 405N in the same assay. If the AC domain is needed only for trans-action, then NAB-Dna2 Δ 248N-AC or NAB-Dna2 Δ 405N should behave similarly as NAB-Dna2 Δ 248N.

Reviewer #2:

Remarks to the Author:

This is a very nice manuscript detailing the interactions and effects of same between RPA and Dna2.

The work features a sophisticated application of multiple state of the art methods that all help to reinforce the main conclusions that RPA affects Dna2 processivity and that the interaction is focused on the N-terminus of RPA. There is more. My comments are generally minor.

The summary is too long by a factor of 2. The main points are not encountered until one has read almost 2/3 of it. This will discourage potential readers.

I assume these are all proteins from yeast, but that should be mentioned somewhere early for general readers.

Line 96. What salt is being referred to? It is not mentioned here and it is hard to know what is meant by "salt" at most points in the manuscript. KCl is mentioned late in the text and NaCl is mentioned in some of the figures. Neither is the physiological salt in cells; e.g., the concentration of Cl ion in yeast is generally maintained at < 1 mM regardless of the NaCl or KCl concentration outside the cells. This should be acknowledged and the word "salt" replaced with the actual salt being used in most places.

Reviewer #3:

Remarks to the Author:

In this is a very interesting manuscript Shen and colleagues combined single-molecule curtains, biochemical approaches and in vivo studies to investigate the interaction between yeast Dna2 DNA helicase and ssDNA binding protein RPA. Importantly, the authors discovered a region at the N-terminus of RPA1 which interacts with Dna2 and allosterically activates Dna2 processive translocation. This interaction with RPA is also important for the Dna2 recruitment to ssDNA under physiological conditions. The reciprocal site (or at least one of the sites), an acidic patch, on Dna2 has also been identified. In addition to Dna2 activation, RPA's more classic role of melting secondary structures of DNA seems important for longer range processivity of Dna2.

Overall, the experiments reported here are carefully designed and expertly executed. The results are convincing and will be important to the field as they will elevate our understanding of the Dna2-RPA.

A few points:

1. In the model the authors present (Fig. S6 and corresponding discussion), the potential role of the RPA-Dna2 interaction in removing RPA from ssDNA during processive DNA degradation seems to be overlooked.

Similarly, it would be informative if the authors discuss how the activation by RPA works in the context of Dna2 activation by other players in long range DSB resection. It has been shown recently that human CtIP activates DNA2 in long range end resection (Ceppi et al 2020 PNAS); would it be reasonable to expect that a similar interaction exists in the yeast system and how would this fit with RPA-Dna2 interaction?

2. The authors discuss RPA dynamics, but it is unclear what do they mean under dynamic nature of RPA. Is it dynamic interactions between RPA and its partners? Microscopic dynamics of the RPA bound to ssDNA and its modulation by interacting proteins (see for example Pokhrel et al 2019 NSMB)? Flexibility of the RPA-ssDNA complex (e.g. Yates et al 2018 Nature Comm)?

3. The authors propose that in the long range nuclease activity of Dna2, both the ability of RPA to activate Dna2 and to melt secondary structures are important. To separate the two influences, could the authors generate a long ssDNA overhang without secondary structures? Alternatively, they could combine their fused NAB-Dna2-delta248N construct with a heterologous RPA? Either of these experiments will strengthen the model.

Our point-by-point response to the Critiques:

We are very pleased that all three reviewers find our work of general interest. We would like to sincerely thank all three referees for their time reviewing our manuscript and their excellent comments to improve our study. As documented below, we strived to address all the comments from the referees and incorporate all their excellent suggestions in the revised manuscript, which, as a result of revision according to the critiques, is stronger than the original version.

REFEREE 1:

This referee noted that “Overall, this study furnishes insights into a hitherto obscure mechanism germane for understanding the processing of HR intermediates and mechanisms of genome maintenance. The analyses have been conducted exceptionally well and the conclusions are supported by the results.”

The referee made a number of suggestions to help us add new mechanistic insight to the story. We are grateful for these suggestions, and below, we document how we have incorporated them into the revised manuscript.

1. The referee asked us to expand the introduction section to better discuss the mechanism of Okazaki fragment and DNA end resection.

Our response: We would like to thank the referee for this suggestion, and we apologize for the lack of an introduction section due to the format for the original submission. We reorganized our manuscript to match the format for *Nature Communication* and to include an introduction section, where the lagging strand maturation and DNA end resection were discussed.

2. The referee asked us to examine the impact of ATP presence on Dna2 processivity in the DNA curtains analysis.

Our response: We have performed this analysis as instructed and compared Dna2-catalyzed digestion of RPA-GFP coated long ssDNA substrate in the absence or presence of 1 mM ATP. The result, included as **Figure S2H**, showed that the presence of ATP has no significant impact on Dna2 processivity in DNA Curtains analysis.

3. Knowing the functional importance of the OB-fold domain in Dna2, the referee asked us to test dna2-2 (dna2-P504S) for ternary complex formation with RPA and ssDNA.

Our response: During the revision process, we generated dna2-P504S construct and purified the mutant protein (**Figure S7A(vi)**). dna2-P504S had a low yield comparable to dna2-Δ501N., Interestingly, purified dna2-P504S is severely defective in the digestion of 5' overhanging DNA regardless of the absence or presence of RPA (**Figure S3E(i)**). dna2-P504S also failed to support the formation of a stable ternary complex (**Figure S3E(ii)**), which is consistent with the role of Dna2 OB-fold in the ternary complex formation.

4. Knowing the N-terminus of RPA is crucial for complex formation, the referee asked us to test a Rfa1-N mutant, such as the rfa1-t11 mutant.

Our response: We would like to thank the referee for pointing this key element out and we generated the Rfa1-NAB-K45E (rfa1-t11) construct and purified the corresponding mutant polypeptide (**Figure S8E**). The Rfa1-NAB-K45E is defective in supporting Dna2's initial cleavage (**Figure 5D(i)** and **(ii)**), but it can partially support the ternary complex formation (**Figure 5D(iii)**). Besides K45E, we had also screened the 15 aromatic residues in the N-terminus of Rfa1 to look for residues important for gating of the single-stranded DNA, which was not included in the original submission and is now included as **Figure 5** and **S5**. Among the 14 single and double aromatic mutants we made, F15A, Y29A, Y55A, Y96A, Y123A/F124A and Y193A each displayed a partial defect in supporting Dna2's initial cleavage, which affirms the importance of the Rfa1-N domain in Dna2 regulation. Importantly, Y193A also displayed a defect in the progressive digestion of ssDNA (**Figure 5B**). In the DNA curtains analysis (**Figure 5C**), RPA-Y193A, while can still recruit Dna2 to the 5' end of ssDNA, failed to support long ssDNA digestion by Dna2. Hence, the linker region between the Rfa1-N and the Rfa1-DBD-A domains likely plays an important role in ssDNA gating.

5. The referee asked us to test NAB-Dna2 Δ 248N-AC or NAB-Dna2 Δ 405N to confirm that the fusion void the *trans* action.

Our response: We have constructed the NAB-Dna2 Δ 405N and purified the recombinant fusion protein. The NAB-dna2 Δ 405N behaves similarly to NAB-dna2 Δ 248N in both the initial cleavage (**Figure S7B(i)** and **(ii)**) and the processive digestion of a 5' overhang (**Figure S7B(iii)**), where it can be stalled by a DNA hairpin as well (**Figure S7B(iv)**). Thus, the fusion of NAB to Dna2 does void the *trans* action mediated by the AC domain.

REFEREE 2:

This referee noted that “This is a very nice manuscript detailing the interactions and effects of same between RPA and Dna2.”

This referee made a few comments to help us strengthen the manuscript and make it more approachable to potential readers. We would like to thank this referee for these comments, which have been incorporated into the revised manuscript as described below.

1. The referee asked us to trim the abstract so that the main point is highlighted.

Our response: We would like to thank this referee for the suggestion, and we apologize for the long summary paragraph in the original submission. We have revised our abstract and trim it to less than 149 words according to the guidelines from *Nature Communications*. The revised version is more succinct and easier for the reader.

2. The referee asked us to specify the organism that we work with early in the manuscript for general readers.

Our response: Again, we would like to thank this referee for the suggestion, and we have specified our system, budding yeast - *Saccharomyces cerevisiae*, in the revised abstract.

3. *The referee asked us to specify the actual salt, viz. NaCl or KCl, used instead of using the general word “salt” since neither is the physiological salt in cells; e.g., the concentration of Cl ion in yeast is generally maintained at < 1 mM regardless of the NaCl or KCl concentration outside the cells. The referee also asked us to acknowledge this.*

Our response: We completely agree with the referee and now have the type of salt used specified in the main text as suggested. We also acknowledged in the revised methods under the “Dna2 nuclease assays” section that neither KCl nor NaCl represents the physiological salt in cells.

REFEREE 3:

This referee noted that “Overall, the experiments reported here are carefully designed and expertly executed. The results are convincing and will be important to the field as they will elevate our understanding of the Dna2-RPA.”

This referee made a few comments to help us improve the manuscript, especially the discussion of our findings. We are truly grateful for these comments and now have them incorporated into the revised manuscript as detailed below.

1. *The referee noted that the potential role of the RPA-Dna2 interaction in removing RPA from ssDNA during processive DNA degradation seems to be overlooked in the discussion, and also suggest we discuss how the other players in DNA end resection, such as CtIP/Sae2, may affect the function of Dna2-RPA pair.*

Our response: We appreciate these suggestions and now have them incorporated into the revised discussion section. Indeed, our data from analyzing the NAB-dna2- Δ 248N fusion did suggest that the *cis* action within the Dna2-RPA ensemble can remove RPA from ssDNA likely through DNA degradation. However, knowing that the fusion protein fails to turnover from long ssDNA, disassembling of the ternary complex upon the completion of the ssDNA digestion may require the *trans* action between the AC motif and the free RPA molecule. We also added a section to discuss the potential impact of other resection factors including CtIP/Sae2 on RPA-Dna2 interaction. Although the reported stimulation of human DNA2 by CtIP is ATP-dependent and likely relies on the motor activity of DNA2, the Sgs1 helicase, which also interacts with Dna2, on the other hand, possesses an acid patch that has been shown to interact with Rfa1 (Hegnauer, EMBO J, 2012). Given the resection defects that we observed with dna2-AC, it will be interesting to further investigate the interplay of AC motifs from Sgs1 and Dna2 with Rfa1-N during DNA end resection.

2. *The referee asked us to elaborate and clarify the RPA dynamics mentioned in the discussion.*

Our response: We apologize for this confusion. Indeed, we were referring to the dynamic interactions between RPA and its partners as this referee pointed out. We have added a sentence to clarify this in the discussion section as “Our study thus provides a new paradigm on recognizing the dynamic nature of RPA, where, through different modes of interaction, RPA can actively

deliver its bound ssDNA to downstream enzymes/factors and coordinate with them to further resolve DNA secondary structures encountered.”

3. To confirm the dual role of RPA in the long-range nuclease activity of Dna2, the referee asked us to examine the digestion of a long ssDNA overhang without secondary structures or combine NAB-dna2- Δ 248N with a heterologous RPA.

Our response: We thank the referee for this suggestion. As shown in **Figure 4D** (first panel), removal of the internal hairpin located 80-90 nt from the labeled 3' end released the pausing at this position observed in **Figure 4C**, which affirmed the role of *trans* action between Dna2 and RPA in overcoming DNA structures. We also tested heterologous RPA with NAB-dna2- Δ 248N fusion as shown below. While free yeast RPA can facilitate the fusion protein to overcome the structural barrier, human RPA or *E.coli* SSB strongly inhibits the NAB-dna2- Δ 248N, which makes the interpretation difficult and we chose not to include this experiment in the revised manuscript.

Figure R1. Digestion of 3'-labeled 140-nt 5'-overhanging ssDNA (5nM) (hairpin-containing) by titrated NAB-dna2- Δ 248N (0nM to 5nM) without or with yeast RPA (200nM), human RPA (200nM) and *E. coli* SSB (200nM).

Reviewers' Comments:

Reviewer #1:

Remarks to the Author:

The authors have done a fabulous job addressing all my points.

The study has been carried out with great care and the results are impactful for understanding the conserved mechanism of DNA break repair in eukaryotes.

This is another splendid contribution by the Niu laboratory and its collaborators. Publication in Nature Communications is highly recommended.

Reviewer #3:

Remarks to the Author:

The authors adequately responded to my previous critiques. The addition of the data requested by all reviewers and the text edits have much improved this already excellent manuscript.